# Effect of the subjective intensity of fatigue and interoception on perceptual regulation and performance during sustained physical activity

**Aaron Greenhouse-Tucknott**[1]*, **Jake B. Butterworth**[1], **James G. Wrightson**[1,2], **Neil A. Harrison**[3,4], **Jeanne Dekerle**[1]

**1** Fatigue and Exercise Laboratory, School of Sport and Health Sciences, University of Brighton, Brighton, United Kingdom, **2** Hotchkiss Brain Institute, Cumming School of Medicine, University of Calgary, Calgary, Canada, **3** Cardiff University Brain Research Imaging Centre (CUBRIC), Cardiff University, Cardiff, United Kingdom, **4** Department of Neuroscience, Brighton and Sussex Medical School, University of Sussex, Brighton, United Kingdom

\* A.Greenhouse-Tucknott@brighton.ac.uk

## Abstract

### Background

The subjective experience of fatigue impairs an individual's ability to sustain physical endurance performance. However, precise understanding of the specific role perceived fatigue plays in the central regulation of performance remains unclear. Here, we examined whether the subjective intensity of a perceived state of fatigue, pre-induced through prior upper body activity, differentially impacted performance and altered perceived effort and affect experienced during a sustained, isometric contraction in lower body. We also explored whether (cardiac) interoception predicted the intensity of experienced perceptual and affective responses and moderated the relationships between constructs during physical activity.

### Methods

Using a repeated-measures study design, thirty male participants completed three experimental conditions, with the intensity of a pre-induced state of fatigue manipulated to evoke moderate (MOD), severe (SEV) and minimal (control; CON) intensity of perceptions prior to performance of the sustained contraction.

### Results

Performance of the sustained contraction was significantly impaired under a perceived state of fatigue, with reductions of 10% and 14% observed in the MOD and SEV conditions, respectively. Performance impairment was accompanied by greater perceived effort and more negative affective valence reported during the contraction. However, effects were limited to comparisons to CON, with no difference evident between the two experimental trials (i.e. MOD vs. SEV). Individuals' awareness of their accuracy in judging resting heartbeats was shown to predict the subjective intensity of fatigue experienced during the endurance

**Data Availability Statement:** The data underlying the results presented in the study are available from https://osf.io/4gpuc/.

**Funding:** The author(s) received no specific funding for this work.

**Competing interests:** The authors have declared that no competing interests exist.

task. However, interoception did not moderate the relationships evident between fatigue and both perceived effort and affective valence.

## Conclusions

A perceived state of fatigue limits endurance performance, influencing both how effortful activity is perceived to be and the affective experience of activity. Though awareness of interoceptive representations of bodily states may be important to the subjective experience of fatigue, interoception does not modulate the relationships between perceived fatigue and other perceptual (i.e. effort) and affective constructs.

## Introduction

Fatigue is a disabling symptom which may be defined across at least two independent, but interactive, attributes: 1) an objective change in cognitive and/or motor performance (i.e. "performance fatigability") and 2) its subjective perception [1, 2]. The perception of effort during physical activity is offered as the cardinal perceptual feature of fatigue [3] and is often used as a surrogate for the subjective component. Indeed, the perception of effort is the cornerstone of prominent psychobiological models of physical endurance performance [4, 5] representing a key factor involved in the regulation of work-rate [6] and volitional task disengagement [7, 8]. However, although related, perceived effort should not be used as a synonym for perceived fatigue since they represent distinct constructs [9, 10]. For example, an important property of perceived fatigue is that it can be experienced at rest or in the absence of overt activity [11], which may be contrasted to effort, which is experienced in reference to some goal-directed action [9, 12–14]. Indeed, it has been demonstrated that perceived fatigue can be disassociated from effort both during and following physical exertion [11, 15]. The conflation of perceived effort with fatigue and an overreliance on the former in the description of the psychophysiological regulation of physical activity [16] has hindered understanding of the specific role of the subjective experience of fatigue on both performance and its regulation.

The absence of a universally accepted definition and consensus method for assessing subjective fatigue continues to be a significant obstacle to its study [17]. Perceived fatigue is conventionally associated with feelings of tiredness, a lack of energy, exhaustion and a desire to rest [18, 19]. Alternative definitions, however, propose a more reflective phenomenological experience associated with a feeling of reduced capacity to cope with experienced demands [11, 14, 20, 21]. The perception of fatigue has been identified with individuals' perception of their capacity to effectively exert control through action; that is, ones' self-efficacy [22]. Self-efficacy defines an individual's task-specific judgment of their capability to execute an action and attain a desired outcome [23]. Accordingly, fatigue and self-efficacy have recently been proposed to be extensionally equivalent [14]. It is well documented that manipulations of self-efficacy influence endurance performance [24–27] through modulating both perceived effort [24, 25; though see 28] and affective experiences [15]. The development of a perceived state of fatigue may similarly limit physical endurance performance by also influencing similar sensory processes [29]. Yet to the best of our knowledge, direct assessment of the relationships between perceptual and affective constructs is currently limited to a small number of principally correlation-based investigations in healthy populations across both physical and cognitive domains [29–31].

Establishment of causal associations requires explicit manipulation of perceived fatigue and examination of its subsequent effects. The perception of fatigue is often quantified as a continuous variable in which perceptions increase across a range of subjective intensities [11]. Currently, it is unclear what the relationship between the intensity of perceived fatigue and its putative perceptual, affective and performance effects should be. That is, it is unclear whether responses to a greater intensity in perceived fatigue are continuous or categorical in nature. It is possible that incremental increases in perceived fatigue may evoke proportional changes in central regulatory (e.g. perceived effort and affective valence) responses and task performance (e.g. time to task failure). Alternatively, an increase in perceived fatigue may evoke some change in these regulatory responses, but further increments in subjective intensity elicit no further changes (of note, similar considerations have recently been posed of the effort-value relationship [32]). Examination of the self-efficacy–endurance performance relationship has indicated that performance may be differentially affected across low, moderate and high perceptions of perceived performance ability, particularly during performance of novel tasks [33]. Applied to perceived fatigue, this may suggest that the experienced, subjective intensity of fatigue may be associated with distinct perceptual and affective effects which subsequently impact performance outcomes.

Interoception, defined as the encoding and representation of signals (e.g. hormonal, immunological, metabolic, thermal, nociceptive, and visceromotor [34]) reporting the physiological condition of the body [35], is a fundamental component of adaptive (allostatic) behaviour [34]. Interoception is assumed to play an important role in the central regulation of physical endurance performance [36, 37], though few studies have directly investigated this relationship. One of the most commonly adopted methods of quantifying individuals' interoceptive ability is the assessment of ones' perception of their own heartbeats. Indeed, individuals who display greater accuracy in detecting resting heartbeats have been reported to cycle at lower work rates during self-paced cycling exercise than those of poorer accuracy, despite reporting the same intensity of perceived fatigue [38]. Interoceptive processing can be decomposed across three dimensions relating to an individual's accuracy (objective precision in monitoring internal bodily sensations), confidence (subjective ability to detect interoceptive sensations) and awareness (a metacognitive measure of the correspondence between objective and subjective assessments) in monitoring interoceptive stimuli [39, 40]. In a heightened state of fatigue, greater attention may be afforded to (unexpected) ascending interoceptive signals [22], which may reflect salient information used in forming momentary perceptions, such as effort [41]. Accordingly, it may be reasonable to assume that an individual's interoceptive ability may influence subjective responses to physical activity and the putative associations between perceived fatigue and other perceptual (and affective) processes, such as effort [29]. Specifically, we propose that interoceptive awareness may influence the relationship between perceived fatigue and other constructs used within performance regulation because both perceived fatigue [22, 42] and performance regulation [43] may involve compatible, higher-order (meta) cognitions.

The primary aim of the present study was therefore to examine the effect of different subjective intensities of fatigue on subsequent physical endurance performance and self-reported measures of affective valence and effort recorded during activity. We adopted a prior exercise paradigm, in which a perceived state of fatigue was pre-induced by an intermittent, upper body exercise, with the effects on lower body endurance performance and perceptual and affective responses examined [29]. We have previously demonstrated that this paradigm enables the effects of a perceived state of fatigue on the performance of a subsequent physical endurance task to be evaluated within an intact system; that is, independent of concurrent neuromuscular deficits typically incurred through protracted physical exertion [29]. This is in

line with previous findings [44–46]. It was hypothesized that lower-body endurance performance would be impaired when the intensity of the perceived state of fatigue was higher, with a more severe state of fatigue associated with greater effort and more negative affective valence. A secondary aim of the present study was to explore the moderating effect of interoceptive awareness on the relationship between perceived fatigue and both perceived effort and affective valence.

## Materials and methods

### Participants and sample size estimation

The size of the effect reported in our previous study investigating the influence of prior, remote motor activity on endurance performance [29] was used to establish a lower bound sample estimate for the present investigation (N = 20). A sequential sampling rule was then adopted to determine the sample size for the present investigation using the composite open adaptive sequential test [47]. Once N = 20, endurance performance was compared between each condition using paired samples *t*-tests. If the comparison fell between $0.01 < p < 0.36$, further participants were recruited. However, if *p* fell $< 0.01$ or $> 0.36$ for all comparisons, no further participants were recruited [47]. In total, 30 healthy males (mean ± *SD*; age: 25 ± 7 years, weight: 75.4 ± 11.7 kg, height: 1.78 ± 0.06 m) participated in the experiment. Participants were naïve to the true purpose of the study, believing that they were simply taking part in an investigation examining the effect of different durations of prior motor activity on endurance performance in a remote muscle group. Prior to enrolment, all participants were screened based on their medical history and reported no cardiovascular, neurological or musculoskeletal disorders, before providing written informed consent. The cohort were active (4155 ± 2501 MET minutes.week$^{-1}$) based on self-reported activity using the International Physical Activity Questionnaire [48]. All participants were instructed to refrain from caffeine, alcohol, and strenuous exercise for 24 hours prior to each testing session. Hand (26 right) and leg (25 right) dominance was determined using the Edinburgh Handedness Inventory [49]. All procedures were approved by the Life, Health and Physical Sciences Cross-School Research Ethics Committee at the University of Brighton (Ref: 2019–2679) with experimental procedures conducted in accordance with the Declaration of Helsinki, except prior registration in a database.

### Experimental design

The study comprised one preliminary session and three experimental sessions. All sessions were conducted at the same time of day (± 2 hrs) and separated by a minimum of 48 hours. The initiation of each session began with a standardized warm-up, consisting of four bilateral handgrip (HG) contractions at 25%, 50%, and 75% of perceived maximal force [50]. Participants were then asked to perform a series of brief (5s) maximal voluntary contractions (MVCs), alternating between hands (one minute separating each contraction). The same procedure was then replicated in the knee extensors (KE).

In the preliminary session, participants were familiarised with all instruments and experimental procedures involved in the main experimental sessions. Anthropometric characteristics (age, height, body mass) were also obtained, physical activity questionnaires completed and interoceptive ability assessed. The experimental sessions comprised two experimental manipulations and a control condition (CON; Fig 1). The experimental manipulations pertained to the subjective intensity of the perceived state of fatigue induced prior to undertaking a physical endurance task. Two subjective intensities, corresponding to moderate (MOD) and severe perceived fatigue (SEV), were induced using a prior intermittent bilateral HG task and assessed using the rating of fatigue scale (RoF). Immediately (~10 s) after the experimental

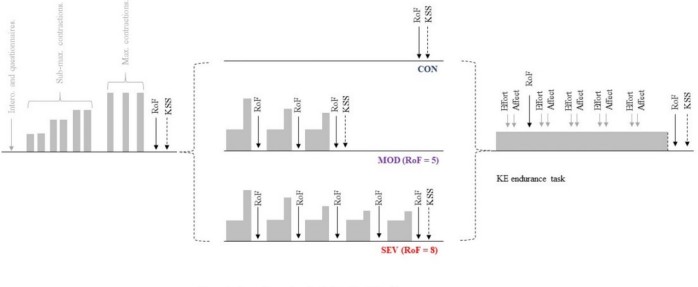

**Fig 1. Schematic representation of the experimental design.** Participants completed three separate conditions. Two involved the performance of an intermittent handgrip (HG) task until ratings of fatigue (RoF) were perceived as moderate (MOD; RoF = 5) and severe (SEV; RoF = 8), representing the experimental manipulation, while the other was a control task (CON) in which no prior exercise was performed. Endurance performance was assessed during a sustained, sub-maximal isometric contraction of the dominant knee extensors (KE), with perceptual (i.e. effort) and affective responses to the task recorded throughout. RoF: Ratings of fatigue; DASS: Depression, Anxiety and Stress scale; KSS: Karolinska Sleepiness Scale; MVC: Maximal voluntary contraction.

manipulations and control intervention, participants performed the KE endurance task. Strong verbal encouragement was provided throughout the HG, but not the KE endurance task to minimize any potential researcher bias from influencing performance on the task. Visual feedback of force production was presented on a monitor, with participants naïve to elapsed time. The experimental sessions were presented in a randomised and counterbalanced order.

**Fatigue-inducing handgrip task and endurance performance task.** Description of the intermittent HG task is outlined elsewhere [29]. In brief, participants initially performed bilateral, submaximal handgrip contractions to a force output equivalent to 30% MVC for 15 s. This was immediately followed by a 5 s MVC and then 10 s of passive rest. This sequence was then repeated continuously. During the rest periods, participants scored their RoF. The HG task was terminated when a specific numerical response was given in each condition. In the development of the RoF scale, descriptors and visual diagrams were orientated in respect to numerical bands, designed to demarcate progressive symptoms across the full perceptual range [11]. Moderately perceived fatigue corresponds to numerical ratings of 4–6, where severely perceived fatigue corresponding to numerical ratings of 7–8. To ensure adequate differentiation between moderate and severe perceived fatigue in the present study, numerical ratings of 5 and 8 were used to separate MOD and SEV conditions, respectively. Once the requisite RoF was achieved the HG task was terminated, with participants naïve to the precise stopping criteria. In CON, participants sat quietly for a duration equivalent to the time taken to perform the HG task to task failure (i.e. force below 30% MVC in the dominant hand for >2 s; determined in the preliminary session), providing a RoF at the end of this period. To ensure the HG task was not terminated after a single contraction in MOD and thus to enable fatigue to development from the intervention, unbeknownst to the participants, all were required to perform a minimum of three contraction sequences in each condition. Participants were instructed to relax their legs throughout the HG task (average EMG amplitude across both muscles and experimental manipulation ranged between 2.1–3.1% max. EMG).

Following the HG (and CON) task, endurance performance was assessed in the dominant KE through a sustained isometric contraction held at 20% MVC until task failure. Task failure was defined as an inability to maintain the target force for >3 s. During the KE endurance task, participants were asked to rate their perceived effort and affective valence every 30 s, with scales presented in a pseudorandomised order. RoF was also assessed 1-minute into the KE

endurance task and upon task failure. Target forces were determined for both the HG and KE endurance task from the highest values recorded during the preliminary session.

## Apparatus and procedures

**Force and electromyography (EMG).** For the performance of the physical tasks, participants were seated, upright on a custom high-backed chair with hip and knee angles set at 90˚ (0˚ = full extension). The upper torso was secured to the back of the chair via two noncompliant cross-over shoulder straps, minimizing extraneous movement of the upper body. Contraction force of the dominant knee extensors (KE) was measured via a calibrated load cell (Model 151/152 S-Beam, Honeywell, Minnesota, USA), secured to the lower leg via a cuff fastened slightly superior (2–4 cm) of the lateral malleoli, which attached to a single channel bridge amplifier (FE221, ADInstruments). Participants' forearms were strapped across the forearm and wrist, in a supinated position (elbow angle set at approximately 70˚; 0˚ = full extension), to a custom table positioned in front of the rig. In each hand, participants held a handgrip force transducer (MLT003/D; ADInstruments, Colorado Springs, CO) with a standardized grip posture. Force was digitized (4 kHz) and analyzed using LabChart v7.0 software (ADInstruments, Oxfordshire, UK). Surface EMG activity was recorded continuously from the dominant *vastus lateralis* (VL), *vastus medialis* (VM), and *rectus femoris* (RF). Pairs of self-adhesive electrodes (Kendall TM H59P, Coviden, Massachusettes, USA), positioned across the respective muscle bellies in accordance to SENIAM guidelines [51]. Reference electrodes was placed on an electrical neural site (i.e. the patella). The skin-electrode interface was prepared by shaving the area, lightly abrading, and cleansing with isopropyl alcohol to minimize electrical resistance. EMG signals were amplified (gain ×1000; PowerLab 26T; ADInstruments, Oxfordshire, UK), digital band-pass filtered (20–2,000 Hz), digitized (4 kHz), recorded, and later analyzed offline (LabChart v7.0).

**Perceived fatigue.** Perceived fatigue was recorded using the RoF scale, with perceived fatigue representing "*a feeling of diminishing capacity to cope with physical or mental stressors, either imagined or real*" [11]. The RoF scale is a single 11-point item, on which perceived fatigue is rated from 0 ('*not fatigued at all*') to 10 ('*total fatigue & exhaustion–nothing left*'). The scale has been shown to have good face validity and high divergent validity from other related, but distinct, perceptual constructs (e.g. perceived effort) [11]. Therefore the scale was adopted to capture the dynamic experience of fatigue arising during the performance of a task that may be missed with the use of other multi-item instruments [11]. Standardised instructions were presented to participants as per the authors recommendations, except for examples that referenced feelings associated with sleepiness or experiences upon waking, to clearly distinguish perceived fatigue from feelings associated with sleepiness [52]. These examples were omitted and/or replaced with ones relating to the subjective experience of physical exhaustion in response to intense exercise. The distinction between the feeling of fatigue and perceived effort was also emphasised.

To assess for the presence of potential confounds on the interpretation of subjective ratings of fatigue, related perceptual and affective states (e.g. sleepiness, depression, etc.) were also assessed during each session. Participants completed the short-form of the depression, anxiety and stress scale (DASS) [53] at the beginning of each session. The short-form of the DASS is a 21-item instrument designed to capture retrospective symptoms experienced over the last week. Each item is rated using a 4-point Likert scale (0 = '*Did not apply to me at all*', 1 = '*Applied to me to some degree, or some of the time*', 2 = '*Applied to me a considerable degree, or a good part of the time*', and 3 = '*Applied to me very much, or most of the time*'). Depression, anxiety and stress scores were obtained by summing the responses provided for the 7 items of

that construct, and then doubling the summed score. In addition, sleepiness, defined here as the momentary feeling of ones' propensity to sleep, was assessed throughout each condition using the Karolinska sleepiness scale (KSS) [54]. The modified KSS is a 9-point scale, ranging from 1 (*'Extremely alert'*) to 9 (*'Extremely sleepy–Fighting sleep'*). Participants were asked to rate their degree of sleepiness concomitantly with their RoF (i.e. before the HG (and CON) task, immediately after the HG (and CON) task and immediately after the KE endurance task).

**Perceived effort.**   Perceived effort was defined as "*the conscious sensation of how hard, heavy and strenuous exercise is*" [55]. Effort was assessed using a category-ratio scale (Borg CR10) [56]. Experiential anchors were used to anchor the scale, with 0 representing *no effort* and 10 representing the level of effort exerted during the strongest contraction an individual had previously experienced (*max effort*). Participants were provided with general and complementary instructions as per the recommendations of the scale's creator [57]. The instructions provided explicitly stated that ratings of perceived effort should be independent of other somatic sensations, including feelings of pain, which may emerge from separate neural processes [58, 59].

**Affective valence.**   Changes in affective valence were assessed using the Feeling Scale (FS) [60]. The FS is an 11-point (+5 to -5) scale which captures the pleasurable (positive integers) and unpleasurable (negative integers) dimension of emotional states, either side feelings of neutral affect (0). Descriptors are provided at all odd integers. Standard instructions were given to all participants prior to experimentation [60]. Extreme ends of the scale were anchored based on experiential factors, with +5 representing individuals' most pleasant experience during previous physical activity and −5 their most unpleasant experience.

**Interoception assessment.**   Cardiac interoception was assessed using a heartbeat discrimination (HBD) task [61]. The HBD is an interoceptive-exteroceptive discrimination task in which participants are asked to judge whether audio tones (440 Hz; 100ms) are played either 'synchronously' or 'asynchronously' with their own heartbeat. Heartbeats were measured through pulse oximetry of the non-dominant index finger (8000s and Xpod, Nonin, Minnesota, USA). Under asynchronous conditions, a 300 ms delay was inserted resulting in tones played approximately 250 ms and 550 ms after the estimated R-wave in the synchronous and asynchronous trials, respectively. The two intervals correspond to maximum and minimum judgements of synchronicity [62]. Each trial consisted of 10 audio tones. Participants were provided with 3 practice trials, before completing 40 test trials, split evenly between synchronous and asynchronous conditions. An accumulation of 40 trials has been recommended as the minimum number of trials required for the HBD task, since reliability and power may be compromised with lower trial numbers [63]. After each trial, participants were asked to rate how confident they were with their response on a 10-cm VAS, between extremes labelled "*0 = total guess/no heartbeat awareness*" and "*10 = complete confidence/full perception of heartbeat*". During the HBD task, participants sat quietly, with their arms and hands supported by cushions on a table directly in front of them, in supinated position. The HBD task was run using a custom script created in MATLAB (MathsWorks, Natick, Massachusetts, USA). Participants were instructed to concentrate on the 'feeling' of their heartbeat wherever they may feel it in their body.

## Data analysis

MVC force was defined as the greatest 500 ms average recorded during maximal contractions. The percentage change in HG MVC force during the HG task was quantified from baseline MVC (i.e. following the warm-up at the beginning of each session) to the last MVC of the HG task. Maximal root mean square ($RMS_{Max}$) was obtained for each quadricep muscle using a

500 ms windowed centred around the highest recorded KE force at baseline. During the KE endurance task, RMS was calculated using consecutive 15 s sampling windows and normalised to RMS$_{Max}$.

During the HG and KE tasks, rate of change in perceived effort and affective valence was derived from the difference between the first and the last response over KE performance time. To compare the average rate of change in perceived fatigue between the two experimental manipulations, an estimate of perceived fatigue accumulated per contraction sequence was derived from the difference between the baseline perceptual intensity and the intensity at the end of the HG task, divided by the number of contraction sequences completed.

Individual interoceptive awareness was assessed based on the three-dimension model proposed by Garfinkel *et al.* (37). Interoceptive awareness was determined using receiver operating characteristic (ROC) curve analysis [64], assessing how well interoceptive accuracy (i.e. proportion of correct judgements) during the HBD task is predicted by interoceptive confidence (assessed via VAS). The association between hit rate (i.e. the proportion of correct heartbeat identifications) and false alarm rate (i.e. the proportion of incorrect heartbeat identifications) were plotted for each detection threshold, with the area under the curve providing a measure of interoception (meta)awareness [40].

## Statistical analysis

Statistical analysis was performed using Jamovi (*v.* 1.6.6) [65], unless stated otherwise. Assumptions of normal distribution were assessed using model residuals through a combination of normal Q–Q plots and Shapiro–Wilk tests. A one-way repeated measures ANOVA was used to assess MVC force between conditions at the beginning of each session. Within the HG task, the relative (%) change in maximal force was assessed between hands and experimental manipulation (MOD *vs.* SEV) using a two-way repeated measures ANOVA. The effect of condition (CON *vs.* MOD *vs.* SEV) on KE endurance performance (i.e. time to task failure; TTF) was assessed using a one-way repeated measures ANOVA. Analysis of RoF was split across each experimental trial: (1) RoF at the beginning of each session was compared between conditions using a Friedman's ANOVA; (2) the difference between recorded and targeted RoF was then tested using one-sample Wilcoxon signed rank tests; (3) the response to the KE endurance task was tested using a two-way (3 x 2) repeated measures ANOVA to assess effects of condition and time (one minute into the KE contraction and upon task failure). Sphericity was assessed using Maulchy's test. Greenhouse-Geisser and Bonferroni corrections were applied where necessary. DASS and KSS were compared between conditions at each respective point of assessment using a Friedman's ANOVA.

A linear mixed model (LMM) was fitted to assess perceived effort and affective valence reported across the endurance task using the *GAMLj* module in Jamovi [66]. The LMM enables effective handling of missing or unbalance data, which was evident in the present study due to differences in endurance performance between individuals. Perceived effort and affective valence were included as dependent variables in separate models. The effects of condition, time and their interaction were entered into the model as fixed effects. The model residuals for affective valence violated the assumptions of the LMM. This was largely driven by one participant who consistently reported the same numerical response throughout the SEV trial which extended >300 s. We report the analysis with this participant removed (n = 29), which rectified the violated assumption. It should be noted, that the conclusions drawn from the best model (see below) largely hold when the whole sample is included (despite the violation of the assumption of normality). The modelling of random effects was initially compared across two models: one in which random intercepts (grand mean) were included across participants and

one in which random intercepts and random slopes for the effect of time varied across participants. Using the Akaike information criterion (AIC) to assess model quality, modelling of effort was best reflected by the inclusion of random intercepts only (random intercepts: 1739.700; random intercepts + random slopes for time: 1774.698), while affective valence was best modelled including both random intercepts and random slopes for the effect of time (random intercepts: 2020.738; random intercepts + random slopes for time: 1910.484). As our purpose was to assess the effect of our experimental manipulation on the perceived effort and affective valence experienced during the KE endurance task, the model was generated only from time points in which recordings in all three experimental conditions were obtained. This led to the exclusion of 12 data points (0.019% of total data recorded), which were taken from the CON and MOD conditions across six participants. F-tests were computed for the fixed effects using Satterthwaite approximation of the degrees of freedom. Analysis of the fixed effect of condition was coded using a Helmert contrast, in which two effects were examined: 1) the difference between CON and the mean across MOD and SEV, 2) the difference between means in MOD and SEV. The effect of time was assessed using a polynomial contrast. Due to issues with the signal-to-noise ratio, five participants were excluded from each of the analyses performed on the EMG response across the different KE muscles. Residuals of the EMG response demonstrated a right-ward skew and thus deviation from the assumption of normality. We therefore adopted a robust approach to the estimation of the mixed effects [67] in *R* [68]. This robust approach attempts to place limits (or bounds) on the influence of sources of error within the random effects structure [67]. Data is presented in the supporting information.

Within-participant, repeated measures correlations were used to assess the relationships between perceptual (i.e. effort and fatigue) and affective responses recorded during the endurance task (i.e. at 1 minute, plus the rate of change across the full task for both effort and affect) and time to task failure using the *rmcorr* package in *R* [69]. Following the exploratory evaluation of relationships in our previous study [29] we applied a more conservative Bonferroni-adjustment to control for multiple comparisons in the present study.

Finally, exploratory analyses were performed to assess the influence of interoception on perceptual and affective constructs. First, LMM were used to examine whether interoceptive dimensions predicted RoF, perceived effort and affective valence during the initial stages (i.e. at 1 minute) of the endurance task across conditions, with intercepts entered as random effects across participants. Next, the moderation effects of dimensions of interoception were examined on the ability of RoF to predict perceived effort/affective valence. This was performed again using a LMM with the intercept entered as a random effect across participants.

Data for parametric analyses are reported as mean±*SD*, while non-parametric analyses are reported as median (*Mdn*) plus interquartile range (*IQR*), unless otherwise stated. Effect sizes for main effects are presented as partial eta squared ($\eta_p^2$), while the pairwise comparison of paired mean differences are presented as Cohen's $d_{av}$ [70] for parametric analyses. The null hypothesis was rejected at an α-level of .05.

## Results

### Preliminary analysis: Force generating capacity and negative emotional states

Initial assessments were performed to assess whether participants differed in both their functional (i.e. MVC force) and emotional state (i.e. depression, anxiety and stress) at the start of each session. The results demonstrated no statistical difference in MVC force (all *p*>0.05;

) or retrospective recall of depressive, anxiety and stress-related symptoms preceding each session ().

## Assessment of experimental manipulation: Handgrip task performance and perceived fatigue

A greater number of contraction sequences were required to induce a RoF of 8 (SEV) *vs.* 5 (MOD) (Table 1; $p<0.001$). A greater relative reduction in HG force (-47.8 ± 11.4%) was also observed in SEV compared to MOD (-38.1 ± 9.4%; $F_{(1,29)} = 29.62$, $p<0.001$, $\eta_p^2 = 0.505$). This reduction in force did not differ between the dominant and non-dominant hand ($F_{(1,29)} = 0.03$, $p = 0.861$, $\eta_p^2 = 0.001$) and there was no significant condition by hand interaction ($F_{(1,29)} = 1.22$, $p = 0.278$, $\eta_p^2 = 0.040$).

Mean RoF at the beginning of each session was low (RoF <2; <"*a little fatigued*") and was not statistically different between conditions (Table 1; $\chi^2_{(2)} = 0.756$, $p = 0.685$). During the HG task, RoF increased progressively, with the average rate of change in RoF not statistically different between MOD and SEV (Table 1; $t_{(29)} = 1.35$, $p = 0.187$, $d_{av} = 0.21$). In SEV, final ratings (mean = 8.0, *Mdn* = 8, range = 8–9) were not different from the target value (RoF = 8; $p>0.999$). All participants, except one (RoF = 9), had the HG test terminated after reporting an RoF of 8. Conversely, the RoF in MOD (mean = 5.4, *Mdn* = 5, range = 5–6) did differ from the target value (RoF = 5; $p = 0.001$). Eleven participants reported an RoF of 6 at the end of the HG. Despite some exceeding the specified value, the RoF did not exceed the moderate intensity band (>RoF 6; $p>0.999$). In CON, RoF remained low (mean = 1.4, *Mdn* = 1, range = 0–3) and did not exceed a RoF of 3 ($p>0.999$). The change in RoF with experimental manipulation was not accompanied by changes in perceived sleepiness at the end of the HG task (S2 Table) demonstrating participants' ability to clearly distinguish between constructs.

## Knee extensor (KE) endurance performance: Time to task failure

Participants' ability to sustain the KE endurance task was significantly affected by the experimental manipulation (Fig 2; $F_{(2,58)} = 11.1$, $p<0.001$, $\eta_p^2 = 0.278$), with a shorter performance in MOD (219 ± 68 s, -9.5 ± 19.1%; $t_{(29)} = 3.11$, $p = 0.013$, $d_{av} = 0.37$) and SEV (206 ± 59 s, -13.7 ± 17.2%; $t_{(29)} = 4.29$, $p<0.001$, $d_{av} = 0.57$) compared to CON (246 ± 79 s). Endurance performance between MOD and SEV was not statistically different ($t_{(29)} = 1.62$, $p = 0.348$, $d_{av} = 0.20$).

**Table 1. RoF at the beginning of each session and the physical and perceptual responses to the HG task performed in the experimental manipulations.**

| | CON | MOD | SEV |
|---|---|---|---|
| RoF (A.U.) | 1.57 ± 1.04 | 1.53 ± 1.17 | 1.37 ± 0.89 |
| | (1.0, 1.75) | (1.5, 1.0) | (1.0, 1.0) |
| HG Task Performance | | | |
| *Number of Contractions* | | 4.50 ± 1.61 | 8.43 ± 2.65 * |
| | | (4.5, 2.75) | (8.0, 3.75) |
| *Relative change in dominant HG force (%)* | | -38.6 ± 8.72 | -47.5 ± 12.6 * |
| *Relative change in non-dominant HG force (%)* | | -37.5 ± 10.1 | -48.2 ± 10.2 * |
| *Average rate of change in RoF per contraction (A.U.contraction$^{-1}$)* | | 0.92 ± 0.32 | 0.86 ± 0.25 |

Data presented as mean ± *SD*. *Mdn* and *IQR* reported for select variables in parentheses. CON: control; MOD: moderate RoF; SEV: severe RoF.

*: Significantly differences between MOD and SEV ($p <0.05$)

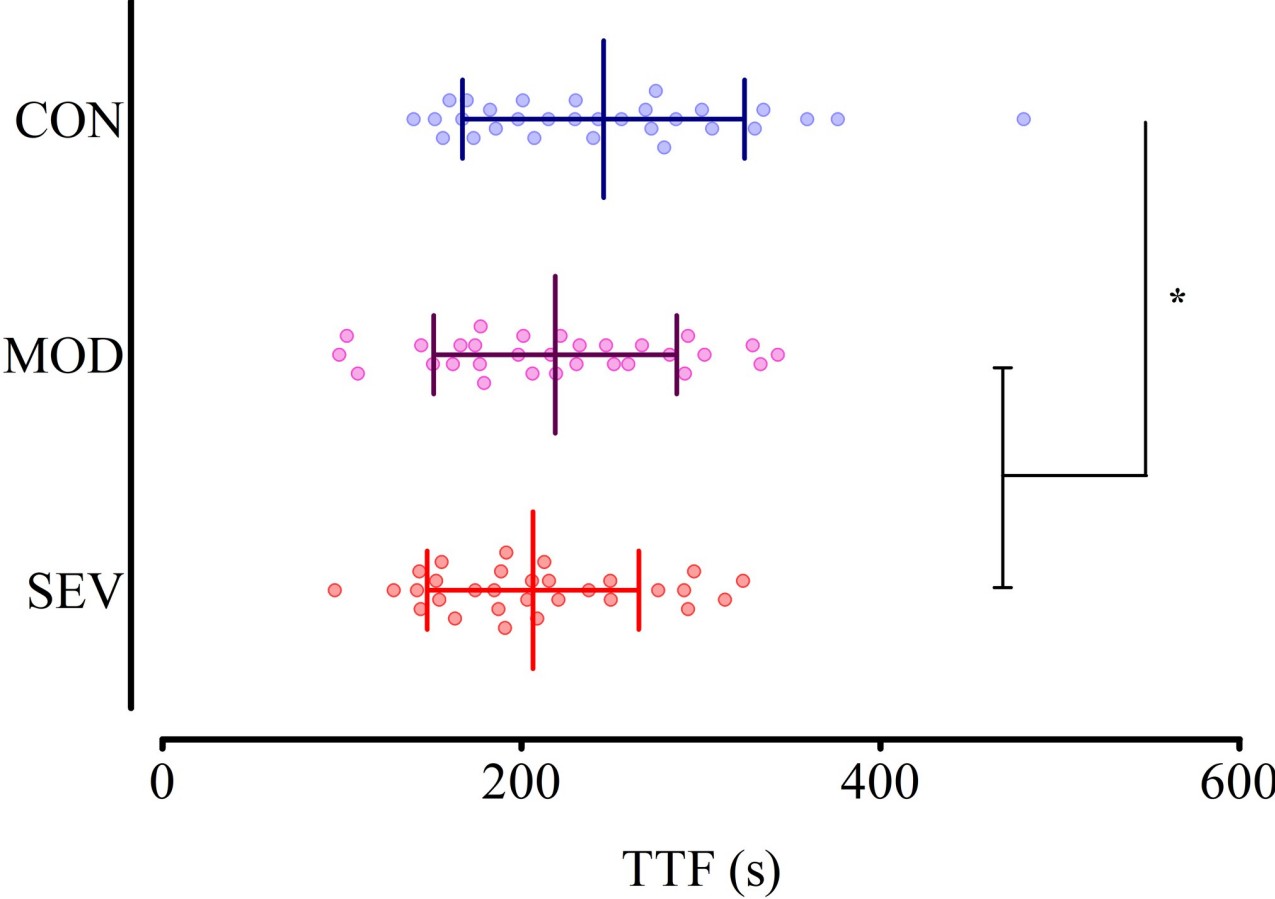

**Fig 2. The effect of the subjective intensity of the pre-induced state of fatigue on endurance performance.** Knee extensor endurance performance presented for the control (CON; blue), moderate RoF (MOD; purple) and severe RoF conditions (SEV; red). Lines and error bars indicate condition means and standard deviations, respectively. *: Time to task failure (TTF) significantly different in MOD and SEV compared to CON (p<0.01).

### Perceptual and affective responses to endurance performance: Perceived fatigue, effort and affect

RoF recorded during and after the KE endurance task are presented in Fig 3. Significant main effects for condition ($F_{(2,58)}$ = 18.14, $p$<0.001, $\eta_p^2$ = 0.385) and time ($F_{(1,29)}$ = 176.64, $p$<0.001, $\eta_p^2$ = 0.859) were found, with a significant interaction between condition and time evident ($F_{(2,58)}$ = 9.51, $p$<0.001, $\eta_p^2$ = 0.247). Follow-up analysis of the interaction effect demonstrated RoF during the initial minute of the KE endurance task were elevated in MOD ($t_{(29)}$ = -5.71, $p$<0.001, $d_{av}$ = 1.05) and SEV ($t_{(29)}$ = -5.28, $p$<0.001, $d_{av}$ = 1.18) compared to CON. However, at this point, RoF between the two experimental manipulations was not statistically different ($t_{(29)}$<0.01, $p$>0.999, $d_{av}$<0.01). In MOD, the average response remained similar to that induced by the HG task (Mean: 5.4; Mdn: 5) however, in SEV, RoF appeared to systematically decrease from the RoF reported at the end of the HG task (Mean: 5.4; Mdn: 5). Both sets display substantial variation around their mean (MOD: $SD$ ± 1.7, range = 2–9; SEV: $SD$ ± 1.4, range = 3–8). At task failure, there was no statistical difference in RoF between conditions (CON: 7.6 ± 1.2; MOD: 8.0 ± 1.0; SEV: 8.3 ± 1.2; CON *vs.* MOD: $t_{(29)}$ = -1.72, $p$>0.999, $d_{av}$ = 0.32; CON *vs.* SEV: $t_{(29)}$ = -2.75, $p$ = 0.152, $d_{av}$ = 0.60; MOD *vs.* SEV: $t_{(29)}$ = -1.51, $p$>0.999, $d_{av}$ = 0.27). A main effect of condition on sleepiness was indicated at the end of the KE

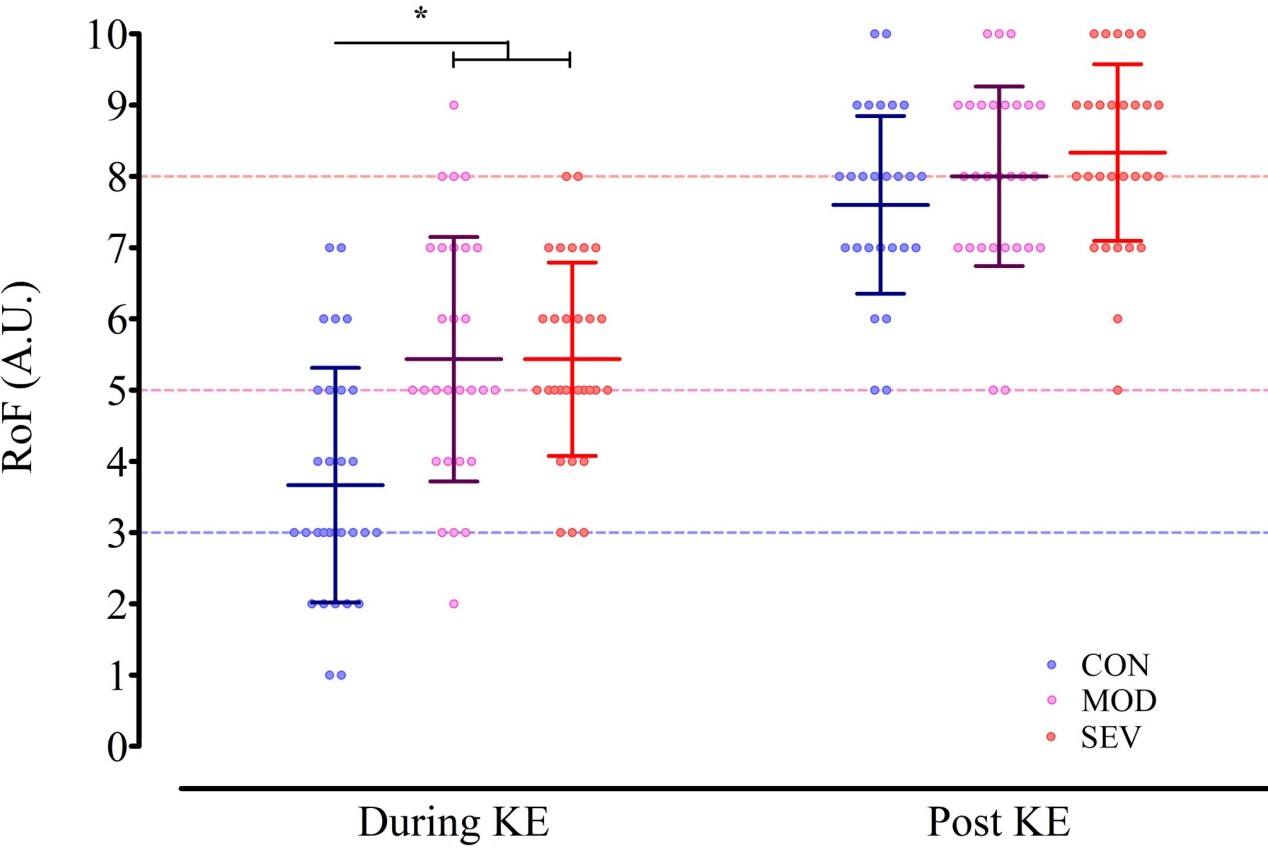

**Fig 3. Effect of the experimental conditions on perceived fatigue.** Rating of Fatigue (RoF) reported both during and post the knee extensor (KE) endurance task is presented for the control (CON; blue), moderate RoF (MOD; purple) and severe RoF conditions (SEV; red). Circles represent individual ratings in each condition. Lines and error bars indicate condition means and standard deviations, respectively. Dashed guidelines represent the target RoF prior to the start of the endurance task in each condition. *: RoF significantly different in MOD and SEV compared to CON (p<0.01).

endurance task (S2 Table), however follow-up pairwise comparisons failed to demonstrate significant effects between conditions (all p >0.05).

Description of the estimated fixed and random effects for the analysis of perceived effort and affective valence recorded during the KE endurance task are presented in S3 Table. Individual and mean responses across each condition for perceived effort and affective valence are presented in Figs 4 and 5, respectively. Main effects of condition ($F_{(2, 556)} = 48.62$, $p<0.001$, $\eta_p^2 = 0.149$) and time ($F_{(9, 557)} = 357.12$, $p<0.001$, $\eta_p^2 = 0.852$) were evident for perceived effort during the KE endurance task, however, the interaction between condition and time was not statistically significant ($F_{(2, 556)} = 0.55$, $p = 0.935$, $\eta_p^2 = 0.002$). There was a linear trend for the effect of time, which flattened out towards the end of the task (quadratic trend; S3 Table). Contrasts between conditions indicated that the average effort response in the combined experimental manipulations (MOD, SEV) was greater than that recorded in CON. The difference between the MOD and SEV however, was not statistically significant (S3 Table). Affective valence decreased over time during the KE endurance task ($F_{(9, 67.2)} = 28.20$, $p<0.001$, $\eta_p^2 = 0.791$). Similar to perceived effort, there was a trend for a linear decrease which tended to flatten out across time (quadratic trend; S3 Table). There was also a main effect of condition ($F_{(2, 388.2)} = 19.06$, $p <0.001$, $\eta_p^2 = 0.089$). Contrasts between conditions demonstrated greater affective valence during CON compared to the average response over the combined

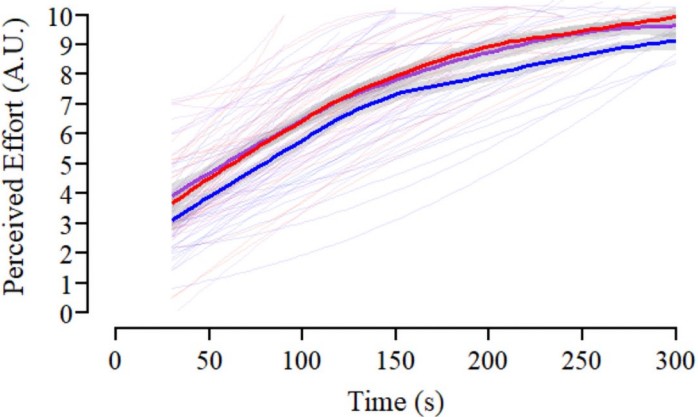

**Fig 4. Effect of experimental conditions on perceived effort during the endurance task.** Profile of subjective perceived effort responses across the knee extensor (KE) endurance task within the control (CON; blue), moderate RoF (MOD; purple) and severe RoF conditions (SEV; red). Individual responses across time are displayed by smoothed thin lines within each condition. Thick lines represent the change in mean scores over time in each condition, with grey boarders representing 95% confidence intervals.

experimental manipulations (MOD, SEV). Affective valence was not different between the MOD and SEV conditions (S3 Table; Fig 5). The interaction between condition and time was not statistically significant ($F_{(18, 455.5)} = 0.70$, $p = 0.813$, $\eta_p^2 = 0.027$).

## Regulation of endurance performance: Relationships between perceptual responses and time to task failure

Correlations were evident between time to task failure and perceived effort after the first minute ($r_{rm} = -0.46$ [95% CI: $-0.64$ to $-0.23$], $p<0.001$), the rate of change in effort ($r_{rm} = -0.41$ [95% CI: $-0.60$ to $-0.17$], $p = 0.005$), the rate of change in affective valence ($r_{rm} = 0.38$ [95% CI: $0.13$ to $0.58$], $p = 0.014$) and perceived fatigue after the first minute ($r_{rm} = -0.54$ [95% CI:

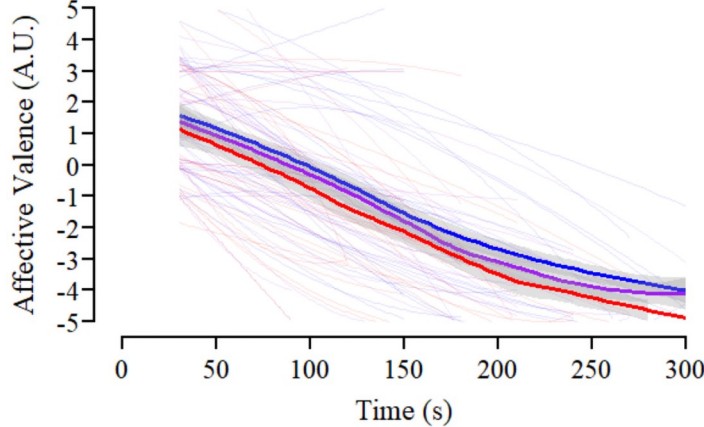

**Fig 5. Effect of experimental conditions on affective valence during the endurance task.** Profile of reported affective valence across the knee extensor (KE) endurance task within the control (CON; blue), moderate RoF (MOD; purple) and severe RoF conditions (SEV; red). Individual responses across time are displayed by smoothed thin lines within each condition. Thick lines represent the change in mean scores over time in each condition, with grey boarders representing 95% confidence intervals.

**Table 2. Moderating effect of the dimensions of interoception on the relationship between perceived fatigue and effort.**

| Fixed Effects | Interoceptive Accuracy | | | | | Interoceptive Confidence | | | | | Interoceptive Awareness | | | | |
|---|---|---|---|---|---|---|---|---|---|---|---|---|---|---|---|
| | Estimates | 95% CI | | $t$ | $p$ | Estimates | 95% CI | | $t$ | $p$ | Estimates | 95% CI | | $t$ | $p$ |
| | | Lower | Upper | | | | Lower | Upper | | | | Lower | Upper | | |
| *Intercept* | 4.58 | 4.16 | 5.01 | 21.14 | <0.001 | 4.60 | 4.17 | 5.02 | 21.35 | <0.001 | 4.59 | 4.18 | 5.00 | 21.89 | <0.001 |
| *RoF* | 0.34 | 0.23 | 0.44 | 6.21 | <0.001 | 0.35 | 0.24 | 0.46 | 6.36 | <0.001 | 0.33 | 0.23 | 0.44 | 6.11 | <0.001 |
| *Interoceptive Dimension* | 1.26 | -1.86 | 4.38 | 0.79 | 0.436 | 0.12 | -0.19 | 0.42 | 0.76 | 0.456 | -2.99 | -6.82 | 0.84 | -1.53 | 0.137 |
| *Interaction* | 0.40 | -0.45 | 1.24 | 0.92 | 0.359 | 0.05 | -0.04 | 0.14 | 1.03 | 0.307 | -0.08 | -1.10 | 0.94 | -0.15 | 0.879 |
| **Random Effect** | *Participant (Intercept)* | | | | | | | | | | | | | | |
| $\sigma^2$ | 0.47 | | | | | 0.47 | | | | | 0.48 | | | | |
| $\tau_{00participant}$ | 1.25 | | | | | 1.24 | | | | | 1.14 | | | | |
| ICC | 0.73 | | | | | 0.73 | | | | | 0.71 | | | | |

Interaction: the interaction effect between RoF and the respective interoceptive dimension, RoF: Rating of Fatigue, $\sigma^2$: residual variance, $\tau_{00participant}$: individual variance, ICC: intraclass correlation coefficient.

−0.70 to −0.32], $p$<0.001). Time to task failure was not related to affective valence reported after the first minute across conditions ($r_{rm}$ = 0.20 [95% CI: −0.06 to 0.44], $p$ = 0.166).

## Exploring the influence of interoception on perceptual and affective responses

We explored the potential influence of interoception on the perceptual and affective responses to the endurance task and its potential moderating effect on the relationships between constructs. First, we examined whether interoception predicted perceptual and affective responses recorded during the endurance task. Though interoceptive accuracy (estimate = 1.57, [95% CI: −1.66 to 4.81], $p$ = 0.348) and confidence (estimate = -0.02, [95% CI: −0.34 to 0.31], $p$ = 0.920) did not, interoceptive awareness was shown to predict RoF reported during the endurance tasks (estimate = -4.18, [95% CI: −8.08 to -0.28], $p$ = 0.048). That is, individuals with greater awareness of resting heartbeats reported lower RoF during the endurance task. None of the dimensions of interoception predicted perceived effort or affective valence recorded at the same time point during the task (Tables 2 and 3).

We speculated that individuals' meta-awareness of interoceptive signals may be particularly important to the relationship between perceived fatigue and effort/affective valence. There was no significant interaction effect between RoF and interoceptive awareness on the prediction of perceived effort (Table 2). This was the same when both interoceptive accuracy and interoceptive confidence were used as the moderator variable (Table 2). Moreover, the same effects also extended to the prediction of affective valence (Table 3).

## Discussion

The present study demonstrated that both moderate and severe intensity of perceived state of fatigue impaired subsequent performance, exacerbating perceived effort and reduced affective valence during the endurance task. However, these effects did not appear to be sensitive to the intensity of perceived state of fatigue, as evidenced by the absence of differences between the two experimental manipulations (i.e. MOD *vs*. SEV). A secondary aim of the study was to explore the potential influence of interoception on the perceptual and affective responses to physical activity and the relationships between constructs. Derived interoceptive awareness

**Table 3. Moderating effect of the dimensions of interoception on the relationship between perceived fatigue and the affective valence.**

| Fixed Effects | Interoceptive Accuracy | | | | | Interoceptive Confidence | | | | | Interoceptive Awareness | | | | |
|---|---|---|---|---|---|---|---|---|---|---|---|---|---|---|---|
| | Estimates | 95% CI | | t | p | Estimates | 95% CI | | t | p | Estimates | 95% CI | | t | p |
| | | Lower | Upper | | | | Lower | Upper | | | | Lower | Upper | | |
| *Intercept* | 0.79 | 0.22 | 1.37 | 2.69 | 0.012 | 0.80 | 0.20 | 1.40 | 2.61 | 0.015 | 0.84 | 0.25 | 1.41 | 2.82 | 0.009 |
| *RoF* | -0.29 | -0.44 | -0.13 | -3.64 | <0.001 | -0.31 | -0.46 | -0.15 | -3.82 | <0.001 | -0.29 | -0.44 | -0.13 | -3.60 | <0.001 |
| *Interoceptive Dimension* | -2.13 | -6.38 | 2.11 | -0.99 | 0.333 | -0.06 | -0.49 | 0.37 | -0.26 | 0.796 | 1.64 | -3.77 | 7.05 | 0.60 | 0.557 |
| *Interaction* | 0.29 | -0.94 | 1.53 | 0.46 | 0.646 | -0.07 | -0.21 | 0.06 | -1.12 | 0.266 | 0.72 | -0.77 | 2.21 | 0.95 | 0.346 |
| **Random Effect** | *Participant (Intercept)* | | | | | | | | | | | | | | |
| $\sigma^2$ | 1.02 | | | | | 0.99 | | | | | 1.02 | | | | |
| $\tau_{00participant}$ | 2.24 | | | | | 2.47 | | | | | 2.26 | | | | |
| ICC | 0.69 | | | | | 0.71 | | | | | 0.69 | | | | |

Interaction: the interaction effect between RoF and the respective interoceptive dimension, RoF: Rating of Fatigue, $\sigma^2$: residual variance, $\tau_{00participant}$: individual variance, ICC: intraclass correlation coefficient.

was the only interoceptive dimensions shown to predict the perceived fatigue reported during the endurance task. The present results therefore indicate that perceived fatigue represents an important factor in the central regulation of physical performance and is associated with higher-order (i.e. metacognitive) representations of internal states. However, interoception did not influence relationships between fatigue and either perceived effort or affective valence during physical activity. This may indicate that the relationships between constructs may be independent of intermediatory sensory processing.

A heightened perceived state of fatigue, induced through prior physical activity (i.e. HG), impaired subsequent endurance performance in a remote muscle group (i.e KE), which was associated with an increased perceived effort and reduced affective valence reported during the task. The observed effect for the reduction in performance time in the SEV condition was comparable to that reported in a recent meta-analysis for endurance-based outcomes following prior, remote physical activity [71]. Several mechanisms accounting for the deleterious effect of prior physical activity on endurance performance in a remote muscle group have been proposed, including neural (e.g. inhibition of descending drive to non-activated muscles), biochemical (e.g. migration of accumulated metabolites) and cognitive factors, though the precise mechanism remains unclear [72]. The present findings appeal to the latter, indicating that higher-order cognitions related to fatigue may exert influence on lower sensory processes independent of overt functional changes [29]. These effects appear relatively potent and may emerge with only a moderate subjective symptoms.

Though effort was elevated, affective valence reduced and performance impaired through elevating the state of perceived fatigue, we observed no statistical differences between the moderate- and severe-intensity manipulations (though there did appear to be a small decline in performance in the severe-intensity trial compared to moderate-intensity trial; -3%). On the face of it, the results suggest that the relationship between perceived fatigue and performance may be categorical in nature; one enters a conscious state of (perceived) fatigue in which performance is impaired and perception of task demands altered after reaching some threshold intensity. Further changes in the intensity of perceived fatigue do not elicit further perceptual or performance effects beyond this point. However, on closer inspection it was evident that individual RoF displayed different response profiles between the two experimental manipulations during the endurance task. That is, RoF decreased in the SEV condition while RoF was

maintained in the MOD condition resulting in no statistical difference between experimental manipulations during the initial minute of the KE endurance task. This may have implications for the interpretation of the current findings. On the one hand, it may aid in strengthening the association between perceived fatigue, effort and affective valence (i.e. given the relationship between perceived fatigue and effort/affective valence, similar RoF between the experimental manipulations may explain why effort, affective valence and performance were also not different), but this may come at the expense of limiting interpretation of the perceptual intensity-performance relationship.

According to the motivational control theory of fatigue, the aversive perceptual response to sustained activity (referred to globally as perceived fatigue) signals the engagement of compensatory, high effort control processes to maintain task performance. The subjective experience of fatigue serves to interrupt goal-related cognitions, enabling one to re-evaluate the utility of engaging effortful processes [42, 73]. According to the authors, changing tasks may attenuate this signal and the need to engage such processes [73], thus a reduction in perceived fatigue coinciding with a new task (i.e. the KE endurance task) in the SEV condition may not be entirely surprising. However, fatigue is also expected to persist for a period after the performance of demanding physical [11] and cognitive tasks [74, 75] since it may function to deter repeated utilisation of the same task-related processes immediately after extended use [73]. Accordingly, it might still be expected that residual fatigue should have been elevated following SEV compared to MOD. Several explanations may offer an account for the absence of this finding: 1) the performance of a new task (or removal of the aversive stimulus) in a state of high fatigue may actually energise the deployment of additional resources during the new task which may in turn transiently alleviate symptoms [74]; 2) perceived fatigue may increase disproportionately to the induced impairment to physical capacity. Perceived fatigue therefore serves as a signal of anticipated adverse consequences should action continue [31] and as such, the disparity between the physical challenge and the perceived challenge becomes realigned once the stressor is removed or a new task starts; 3) the challenge to ones' perceived capacity and the need to exert control in pursuit of a goal may involve multiple goal states that extend over different temporal scales [76], which may see the physiological condition of the body evaluated in relation to new goals introduced by engaging in a new task. Future research is required to better understand the acute changes in perceived fatigue with the engagement of new tasks and re-examine the relationship between the subjective intensity of perceived fatigue and subsequent performance whilst accounting for such changes.

In contrast to our previous findings [29], we found a significant relationship between perceived fatigue and task performance. We have previously argued that although perceived fatigue serves to curtail performance, it may do so indirectly through effects on perceived effort and affective valence [29]. This was also taken as indirect evidence of a differentiation of fatigue from the other studied perceptual (i.e. effort) and affective constructs. The observed relationship between the experience of fatigue during the task and task performance in the present study suggests that perceived fatigue may be more directly involved in the central regulation of performance than previously postulated, forming alongside effort and affect, part of the sensory experience that ultimately signals that the consequences of continuing activity are increasingly unattractive [77, 78]. This does not definitively preclude an indirect effect through changing sensory processing [29], and further work is required to better understand the subtle interactions between perceptual and affective experiences in the regulation of physical behaviour. Here, we demonstrate that perceived fatigue predicts how effortful and how pleasurable activity is experienced to be. Like self-efficacy, a state of perceived fatigue may thus reflect a cognitive factor related to beliefs concerning capacity [14], that shapes how much effort is

invested into a task and the aversiveness of ensuing actions [15, 25], which serves to influence physical tolerance.

Lastly, we explored the possibility that one's interoceptive ability influenced the emergence of the subjective experiences during sustained physical activity and moderated the relationships between fatigue and perceived effort/affective valence. Drawing upon recent descriptions of chronic, pathological fatigue [22], we have previously proposed that the subjective symptom of fatigue arising from acute physical exertion may do so as a result of continued detection of challenges that undermines the experience of control over the body [79]. These challenges are underpinned by discrepancies between top-down expectations, or predictions, of internal states and the sensory evidence received from the body. Reduced confidence in held predictions may subsequently result in greater disparity between what the brain predicts the physiological condition of the body to be and its true state, which subsequently alters the processing of error signals, resulting in greater perceived effort [80, 81] and increasingly negative affective states [82]. Under this framework, we provide a theoretical account of not only how the perception of fatigue emerges during acute physical activity, but also why subsequent activity may be perceived to be more effortful and less pleasurable than normal. The subjective experience of fatigue is offered as an experience emerging from higher-order, metacognitive processing [22] and closely aligned to changes in the estimated precision of descending efferent predictions [79]. In line with this proposition, interoceptive awareness (a measure of an individual's meta-awareness of interoceptive signals) predicted RoF during the endurance task. Specifically, the greater one's interoceptive awareness, the lower the subjective experience of fatigue reported during the endurance task across all three trials. The measure of interoceptive awareness used in the present study is cited as a stable representation of 'error awareness', that may be generalisable across interoceptive axes [83]. The results indicate that the greater the awareness of internal body representations, which one may assume suggests greater confidence in interoceptive predictions of internal states, the smaller or less reliable prediction error is believed to be, resulting in an attenuated experience of fatigue. Importantly, this was evident for perceived fatigue only, with neither effort nor affective valence related to any dimension of interoception. This conforms with previous studies examining cardiac interoception on perceived effort [84] and affective valence [85] during physical activity. Indeed, though we were unable to disassociate constructs based on their relationship to performance, this finding indicates that the studied constructs may involve different circuitry, with metacognition playing an important role specifically in the experience of fatigue.

Though we did not find effects of interoception on perceived effort and affective valence, we speculated that interoception, (and particularly interoceptive awareness) would moderate the relationships between perceived fatigue, effort and valence. The expectation was that awareness would be particularly important for the interpretation of prediction error, and thus the conscious experiences of effort and affect when under a state of fatigue. We found no moderating influence of any dimension of interoception of the relationship between fatigue and either effort or valence. Individuals' interoceptive awareness has previously been associated with perceptual and affective experiences associated with prediction error, including symptoms of anxiety [83] and susceptibility to exteroceptive manipulation of self-location [86]. In the case of anxiety, there is some evidence that the affective response may be sustained by maladaptive metacognitive beliefs about worry [87]. Similarly, the present study suggests that the possible meta-cognitive processing involved in the experience of fatigue may sustain heightened perceptions of task effort and reduced affect arising from lower-level processing. These perceptual and affective responses may not necessitate higher-order cognitions for their experience, even under conditions of uncertainty, such as when fatigued. The relationships

between constructs may instead emerge from the reciprocal interactions between levels within a multi-level hierarchical system and highlight the importance of top-down influences.

Some limitations of the present study must be highlighted. We acknowledge that associations between perceptual responses and endurance performance remain limited to correlational analysis. Further study of the structure of these relationships requires more formal analysis, such as mediation analysis. The study assessed the impact of the subjective intensity of perceived fatigue on performance across three separate levels, which we assumed corresponded to distinct intensity boundaries. Even if this assumption is true, the investigated levels may not have been sufficient to interpret the full (perceived) fatigue-performance relationship. Furthermore, at odds with our previous study [29] and others [45, 46, 88], there appeared to be some evidence of an effect of condition on the EMG response in the KE muscles (S4 Table). This may suggest that motor unit activation may have differed between conditions, meaning that the endurance tasks were not performed in equivalent physiological conditions. This may have contributed to altered perception beyond the proposed top-down (meta)cognitive influences [89]. However, it is acknowledged that inferences drawn on central drive from bipolar surface EMG is problematic [90], and we observed large variation between individuals and only moderate explanation of the variance from the model. As such, we believe that interpretation of the present data is difficult and this effect needs to be investigated further. Finally, the impact of the subjective intensity of a perceived state of fatigue on performance was limited to individual capacity to tolerate a sustained contraction of the lower limbs. More nuanced effects in response to the subjective intensity of perceived fatigue may be gained in tasks where participants are able to vary their responses.

## Conclusion

The primary aim of the present study was to examine whether the subjective intensity of a pre-induced state of perceived fatigue affected subsequent physical endurance performance and altered perceptual/affective regulation. The results demonstrated that an elevated state of perceived fatigue impairs subsequent physical endurance performance and affects how effortful and pleasurable the task is perceived to be. Perceived fatigue was shown to be highly sensitive to changing tasks, which hindered understanding of the precise (perceived) fatigue-performance relationship. Nevertheless, higher-order representations of the interoceptive state were shown to predict perceived fatigue reported during physical activity, implicating metacognitive appraisals as an important component of the subjective experience of fatigue. However, awareness of internal representations did not moderate the relationship between perceived fatigue and other perceptual (i.e. effort) and affective responses arising through engaging in physical activity, indicating separate processing supporting individual constructs used in the regulation of physical performance.

## Supporting information

**S1 Table. Maximum voluntary contraction (MVC) force for the (dominant) knee extensors, the dominant and non-dominant handgrip (HG) at the beginning of each experimental session.** Data presented as mean ± SD.
(DOCX)

**S2 Table. Emotional states and perceived sleepiness between conditions.** Data presented as mean ± SD with Mdn and IQR presented in parentheses. DASS: Depression, Anxiety and Stress Scale. KSS: Karolinska Sleepiness Scale.
(DOCX)

**S3 Table. Estimated fixed effects from linear mixed analysis of perceived effort and affect during the KE endurance task.** Contrast 1 represents the contrast between CON and the combined experimental manipulations (MOD + SEV). Contrast 2 represents the contrast between MOD and SEV. σ2: residual variance, ICC: intraclass correlation coefficient, AIC: Akaike information criterion, R2 marginal: variance explained by the fixed effects over the total (expected) variance of the dependent variable, R2 conditional: variance explained by the fixed and random effects over the total (expected) variance of the dependent variable, CON: control; MOD: moderate RoF; SEV: severe RoF. Full fixed and random effects of models not presented for clarity.
(DOCX)

**S4 Table. Estimated fixed and random effects from robust linear mixed analysis of the relative (%) RMS amplitude in the vastus lateralis (VL), vastus medialis (VM) and rectus femoris (RF) during the KE endurance task.** Contrast 1 represents the contrast between CON and the combined experimental manipulations (MOD + SEV). Contrast 2 represents the contrast between MOD and SEV. $\sigma^2$: residual variance, $\tau_{00participant}$: individual variance, $ICC_{(participant)}$: intraclass correlation coefficient, SE: standard error, $R^2$ marginal: variance explained by the fixed effects over the total (expected) variance of the dependent variable, $R^2$ conditional: variance explained by the fixed and random effects over the total (expected) variance of the dependent variable. Full fixed effects not presented for clarity.
(DOCX)

## Acknowledgments

The authors would like to thank Dr. S. Berens for his contribution to valued discussions concerning statistical methods. The authors also like to thank our laboratory technicians for their support during data collection and the volunteers who participated.

## Author Contributions

**Conceptualization:** Aaron Greenhouse-Tucknott, James G. Wrightson, Neil A. Harrison, Jeanne Dekerle.

**Data curation:** Aaron Greenhouse-Tucknott.

**Formal analysis:** Aaron Greenhouse-Tucknott, James G. Wrightson.

**Investigation:** Aaron Greenhouse-Tucknott.

**Methodology:** Aaron Greenhouse-Tucknott, Jake B. Butterworth, James G. Wrightson, Neil A. Harrison, Jeanne Dekerle.

**Supervision:** James G. Wrightson, Neil A. Harrison, Jeanne Dekerle.

**Writing – original draft:** Aaron Greenhouse-Tucknott.

**Writing – review & editing:** Aaron Greenhouse-Tucknott, Jake B. Butterworth, James G. Wrightson, Neil A. Harrison, Jeanne Dekerle.

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
