## [Decision Letter · Decision Letter 0]

2 Jul 2021

PONE-D-21-15950

Effect of the subjective intensity of fatigue and interoceptive awareness on perceptual regulation and performance during sustained physical activity

PLOS ONE

Dear Authors,

Thank you for submitting your manuscript to PLOS ONE. After careful consideration, we feel that it has merit but does not fully meet PLOS ONE’s publication criteria as it currently stands. Therefore, we invite you to submit a revised version of the manuscript that addresses the points raised during the review process.

Dear Authors,

First of all, my apologies for the time taken in reaching a decision.

As you will see below, the reviewers all agree that you should be commended for the quality of your work. They also raised several concerns. I hope these comments will allow you to further improve your manuscript.

I wish you good luck with the revision process

Best wishes

Mathieu Gruet

Please submit your revised manuscript by August 13 2021. If you will need more time than this to complete your revisions, please reply to this message or contact the journal office at plosone@plos.org. Please include the following items when submitting your revised manuscript:

We look forward to receiving your revised manuscript.

Kind regards,

Mathieu Gruet, Ph.D

Academic Editor

PLOS ONE

Journal Requirements:

Reviewers' comments:

Reviewer's Responses to Questions

**Comments to the Author**

1. Is the manuscript technically sound, and do the data support the conclusions?

Reviewer #1: Yes

Reviewer #2: Yes

Reviewer #3: Partly

2. Has the statistical analysis been performed appropriately and rigorously? 

Reviewer #1: Yes

Reviewer #2: Yes

Reviewer #3: I Don't Know

3. Have the authors made all data underlying the findings in their manuscript fully available?

Reviewer #1: Yes

Reviewer #2: No

Reviewer #3: Yes

4. Is the manuscript presented in an intelligible fashion and written in standard English?

Reviewer #1: Yes

Reviewer #2: Yes

Reviewer #3: Yes

5. Review Comments to the Author

Reviewer #1: This article investigated whether subjective intensity of perceived fatigue, pre-induced through prior upper body activity, differentially impacted performance and altered perceptual (effort) and affective responses during a sustained, isometric contraction in lower body. The authors also explored whether interoceptive awareness moderated the relationship between the perception of fatigue and effort.

I want to congratulate the author for the quality of this very rigorous and well-written scientific paper. This article brings interesting data on how perceived fatigue and effort perception could interact and influence endurance performance. I was naïve on the interoceptive awareness concept, and I think that this strengthen the interest of this paper. I only have some minor comments (and sometimes some personal observations that do not require any answer).

Introduction :

Lines 15-16 : perhaps providing a definition of what perception of effort is should strengthen what you say in the first paragraph, i.e. perception of fatigue should be distinguished from perception of effort. I think that in the definition of “perception of fatigue” it should be stated that this state is independent of any previous physical activity.

Line 20: maybe a clear definition of what “self-efficacy” is will make this part clearer for naïve readers

Line 35: give some example when talking about changes in regulatory responses and performance.

Line 54: give example of interoceptive signals

Material and methods:

The quality of the figures is very low, especially the one of the figure 1 (i.e. the experimental design). It is impossible to correctly read this figure, please improve the quality

Line 156: it is not clear for me; ROF was assessed 1 min after the beginning of the endurance exercise, and at the end. That’s it? Or was it also recorded every 30sec from the 1st min of the exercise?

Maybe change the subtitle “rating of fatigue (RoF)” for “perception of fatigue”. It will be clearer, since the next subsection is called “perception of effort”.

Results:

Table 1: what does “A.U.contraction-1” means?

Lines 297 and 299: what are Mdn ? Median I guess. Maybe the mean value of the ROF could be helpful to see how far it was from the 6 values for the MOD condition

The quality of fig 2 to fig 6 is better than the one of fig 1 but could be much more improved

Paragraph 3.3. It could be a good idea to give the exact time (in sec) for KE endurance performance. We can guess it by checking the fig 2, but it worth it to give the exact values in the text

The results section is already long, I am not sure that table 2 is really needed. Maybe it could better fit in the supp data

Figure 5 should have panel, and you need to use these panels in the text. Further, the legend of one of the panel of the fig 5 is written with a very small police, this is illegible.

Title of the part 3.6: interoceptive and not interceptive

Discussion:

Lines 439-442: I guess that this sentence could go in the conclusion rather than in the first paragraph of the discussion

Line 450: while I agree that it was not the aim of this study, it could be good to give some examples for each causes so as to have a general overview of the mechanisms that account for the deleterious effect of prior physical activity on endurance performance in a remote muscle group

Lines 466-473: a personal comment, that is a very interesting paragraph

Line 494: “a new task”; you mean the transfer from handgrip exercise to KE exercise?

Line 515: which ones?

Lines 520-525: personal comment: another experimental session with the HG exercise until failure (or until a maximal RoF score) could have been interested to meet with what you performed in your previous study (ref 25).

Line 573: is there any possibility for people with “low” individual interoception awareness to improve/train this parameter? Because this interoceptive state can moderate the influence of the perception of fatigue in the perception of effort (thus on the endurance performance), it could be useful to give some recommandations on how a subject with low individual interoception awareness could become a subject with high interoception awareness. I am naïve with this concept, so maybe my comment does not make any sense.

Reviewer #2: Firstly, my apologies for my tardiness in returning my review! Secondly, I just want to say I am continually impressed by the great work being completed by your research group and I’m honoured to be afforded the opportunity to review and provide feedback on it. Below I’ve made notes from my reading of the manuscript. Most are fairly minor to be honest or just annoying quibbles about terminology which are probably to be expected from me. Others are just some of my thoughts and not necessarily in need of being addressed in this manuscript specifically. Lastly, some comments pertain to the statistical analysis and presentation of data. Given this is an open peer review I would be happy to arrange to call and discuss any points highlighted and to offer assistance with any of the analyses should you feel they are worth running. If not then no problem, however I make a note also regarding the data availability and addressing this would mean that readers can look to run some of the suggested analyses (or others) if they wish (and I might at some point).

Thanks,

James Steele

Comments:

Minor quibble, and to be honest I'm not certain it's really that worth changing throughout but I'll mention it anyway... strictly speaking 'perception of' as opposed to 'perceived' is probably more appropriate. Olivier Massin noted the following in feedback on an early draft of my conceptual definitions paper:

"...is subjective effort the perception of effort of perceived effort? You say both, but they can’t be the same (the perception of a tree is not the perceived tree). I would suggest you go for the later, as I’m not sure whether there are many cases in which the perception of an x of kind K is itself of kind K (and I’m quite sure no effort is a perception, while I fully agree that some effort are perceived or felt)"

As I said, a minor quibble and one probably of bigger concern to analytic philosophy than necessarily in empirical work such as this where people know what your referring to. So no real need to address... but I just find it interesting to make people aware.

Line 16: Sensations of tiredness… I'm not sure I would label these as 'sensations' as it is not clear what stimuli are being sensed, and further that sensation isn't phenomenological, but perceptions, feelings are. Maybe the latter is a better label here.

Line 19: I'm not sure I'd say it's a phenomenological counterpart given that self-efficacy is also phenomenological... this is why I've posed that perception of fatigue and self-efficacy are extensionally equivalent and merely opposite signed transformations of one another. Also, I would perhaps define self-efficacy here considering that it is in different places used by people as a state or trait construct, and also that some people consider a 'general' self-efficacy, whereas Bandura's original work for example defined it more in relation to the specific capacity to perform a specific task i.e. self-efficacy for X.

Line 32: Ordinal... although It's also not clear to me whether ontologically classifying fatigue into discrete ordinal bins makes sense anyway.

Line 43: Sense… As above, interoception might be better called a perception... similarly to exteroception. Both are perceptions that are resultant from sensation of some stimuli.

Interoception: I must confess I have not worked with interoceptive awareness measures, but I am not sure I catch the intuition that interoceptive accuracy in one domain (e.g. heartbeats) might be reflective of a general interoception construct. I am aware there are various multi-dimensional measures available - this one for example, though self-report, seems to be of a more general interoception construct https://www.ncbi.nlm.nih.gov/pmc/articles/PMC6279042/

I guess you could argue that, in the vein of my arguments regarding actual-perception of effort, any psychophysical examination of relations between objective and subjective components of some interoceptive construct that do not fall on the identity could be evidence of inaccuracy in interoceptive awareness. Confidence and awareness are sort of akin to that of the Dunning Kruger effect (nice computational model example here - http://haines-lab.com/post/2021-01-10-modeling-classic-effects-dunning-kruger/).

I guess in a round about way I'm just getting at the point of being quite cautious regarding conclusions about the role of 'interoception' per se in your paper as opposed to the relationship between interoceptive ability regarding perception of heart rate and its relationship to your DVs... a slightly more specific conclusion.

Lines 85-87: Did you conduct a debrief with participants and check whether or not they had discerned the purpose of the study or if the deception was maintained?

Line 102: Can you note the duration (seconds) of the MVCs?

Line 118: It’s not clear how it was blinded... participants were deceived as to the overall aims as noted, but surely participants knew which condition they had experienced each time they came in? Or do you mean to say that they didn’t know until they stopped the HG task at a given RoF? In which case they would have been unblinded during the KE task.

Jamovi: Can you add the jamovi file/code to the supplementary materials along with the data in the format used for analysis so that it can be easily reproduced?

Data: So the online materials in excel format are not particularly easy to navigate. Also, I don’t think all the raw data are there. If possible I would re-organise the data (following for example the guidance here - https://www.tandfonline.com/doi/full/10.1080/00031305.2017.1375989). If you had it in a specific format for analysis then I would include that as a separate file and indicate this clearly.

Line 248: It's not clear why it was necessary to use a mixed effects model here. You note that it is flexible to imbalances and missing data, but as you removed data to ensure it was balanced across participants. Though, I do wonder whether you might benefit from the inclusion of random slopes to your model for time... individuals may differ in the growth of effort/affect over time because of different rates of fatigue for example for the former.

medmod: Can you clarify... was this moderation or mediation that was explored? If just moderation then you could have just explored this with a general linear model in base jamovi. Or preferably a mixed model.

Figures: Given the within participant design, it would be better to present the figures with individual low alpha/transparency lines for each participant across the predictor variables in each figure. I would also present interval estimates i.e. CIs. Actually, I would add this throughout the results.

Figure 2: I can’t see the data for this for example in the online file.

Section 3.4: Interesting, and surprising how quickly the SEV RoF dropped and equalised between conditions (I wonder if this occurred within ~10 seconds between HG and KE, or it occurred during the first min of the KE task). Although, Whittaker et al show drops of say ~1-3 pts on their RoF scale during rest periods (see their figure 6; https://www.sciencedirect.com/science/article/pii/S1050641119300136?via%3Dihub). I guess this might explain the lack of difference in performance and perception of effort during the KE task between MOD and SEV as you note.

Figure 4: I find it quite interesting that perception of effort often follows a linear, or convex function... assuming my definition of actual effort, and linear fatigue (which I think is often the case in tasks like this, at least we've found this to be the case in the lab, or in the model simulated in my pre-print [figure 4(b)] where it is proportional to effort at t-1), actual effort increases with time in a concave function. Interestingly though in a few models I've been exploring to examine effort psychophysics on the same ratio scales for actual and perception, isometric endurance tasks display this convex function over time, yet another model of dynamic tasks (elbow flexion with a dumbell) has shown a function a bit closer to the concave actual effort function. I suspect this is something to do with the scale and perhaps ceiling effects, and perhaps also that isometric tasks are inherently more uncomfortable to perform. Of course, you differentiated effort and discomfort here, as do we in our work. But although we have found in several studies that lighter loads/torques to task failure in resistance type exercise cause greater discomfort compared to heavier in dynamic tasks, the data we have for isometric tasks suggests they are similarly uncomfortable (e.g. https://pubmed.ncbi.nlm.nih.gov/32461833/ and also some data for another study we are writing up). It makes me wonder if, even with instruction, people sometimes still 'taint' their ratings of perception of effort with the discomfort of the task when it is particularly salient such as in isometric tasks. Anyway, just some thoughts to share here... not necessarily anything to address given the aims.

Affect: Similarly, to figure 4, one for affect over time would be of interest to show. Or at least have this data available.

Section 3.5: Have you considered comparing these using something like cocor (http://comparingcorrelations.org/)? Although, It's not clear why you would worry about running these separately by condition. You could employ a mixed model approach for repeated measures correlations and calculate this in a single model e.g. https://www.frontiersin.org/articles/10.3389/fpsyg.2017.00456/full. If you felt that were effects specifically by condition categorically you could use similar methods to those used by Vigotsky et al here (code available in their supplementary materials - https://peerj.com/articles/5071/#supp-3) to employ a mixed effect model including condition as an effect. Though I think you should probably just keep things continuous and just color code the figure to show condition.

Figure 5. I would clarify this as being initial ratings of perception of effort in that panel.

Table 2: Presumably, this is a grand mean intercept for these models? If this is what jamovi does as standard it might be worth clarifying this is the case in the methods.

Section 3.6: As noted, I wonder if a better approach here instead of averaging across MOD and SEV would be to model it all together, along with CON. You could just simply explore it in a model like: effort ~ RoF + Int_awareness + RoF*Int_awareness + (1|Participant). You could also add back in the Condition if you really think that interoceptions moderating effect would be moderated itself by this. I can't think why it would be though.

Lines 412-419: This is also interesting, and the opposite of what I would have expected to be honest given I'd expect those with greater interoceptive awareness to have more accurate perception of fatigue, and thus more accurate perception of effort.

Lack of statistically significant differences: I wonder if it is perhaps worth considering looking at whether there was in fact statistical equivalence between MOD and SEV for rating of perception of effort and performance. You would of course need to determine a justifiable smallest effect size of interest for the difference in performance and effort between the two. This is a bit tough given it will obviously be post hoc. But you could base this on reliability of the constructs. You could do the same for the RoF, though it’s not clear what the reliability is to my knowledge… though you could make a reasonable assumption that it’s probably similar to RPE given the similar construction of the scales in terms of points.

Lines 516-519: Might be worth noting changes i affect with time during other multi-joint tasks to TF e.g. https://journals.humankinetics.com/view/journals/ijspp/16/1/article-p135.xml

Reviewer #3: In this study, the authors aimed at investigating the effect of various intensities of perception of fatigue on subsequent endurance performance. To do so, they performed a prior handgrip task until the participants rated two different intensities of perception of fatigue (moderate vs severe) via a recently validated rating of fatigue scale. Then participants had to perform a submaximal isometric contraction at 20% of their maximal voluntary contraction peak torque. The authors observed a reduced endurance performance following prior physical exertion, with no difference between the moderate and severe fatigue conditions. They concluded that perception of fatigue indirectly limits endurance performance by making a physical task appearing more effortful.

I commend the authors for performing a study focusing on the influence of the subjective aspect of fatigue on subsequent endurance performance. This approach is from my point of view important as fatigue manifests both objectively and subjectively, and to the best of my knowledge, the literature focuses mainly on the objective manifestation of fatigue.

Please find below some comments/questions.

1. The authors used prior physical exertion to increase the perception of fatigue. However, physical exertion does not only induce an increase in perception of fatigue but also induces a decrease in force production capacity of the working muscles. Therefore, both subjective (perception of fatigue) and objective (decrease in MVC) aspects of fatigue were induced by the experimental protocol. How did the authors control for this confound?

2. In line with the previous comments, one important limitation of the protocol is the lack of knee extensor maximal voluntary contraction measurement at the onset of the knee extensor endurance task. Without this measurement, the reader cannot be secured that the prior physical exertion did not reduce the maximal force production capacity of the knee extensors. Such a decrease in maximal force production capacity could alter endurance performance independently of any changes in perception of fatigue.

3. In line with points 1 and 2 previously mentioned, why did the authors decided to use prior physical exertion and no other methodologies known to increase the perception of fatigue without decreasing force production capacity such as sleep deprivation (Temesi et al., 2013) or mental fatigue/prior prolonged mental exertion (Brown et al., 2020)?

Contrary to prior physical exertion, prior mental exertion or sleep deprivation does not alter maximal force production capacity, and this confound could have been better controlled. The manuscript could benefit from a better justification of the choice of the experimental manipulation, as well as a clear discussion on the limits of the approach used.

4. The literature on the interaction between mental fatigue and physical performance extensively monitored changes in perception of fatigue induced by prolonged mental exertion (e.g., see work of Marcora, Roelands, Smith, Pageaux). Surprisingly, the authors did not consider this literature in their manuscript as the mental fatigue literature manipulated perception of fatigue and associated changes in performance.

5. The results indicate that feelings of fatigue did not differ between the moderate and severe fatigue conditions early in the knee extensor endurance task (see figure 3). I understand that the difference was evident at the end of the handgrip task as it was the end task criteria.

Based on this observation, can we really consider that the experimental manipulation was successful? In other words, if early in the subsequent exercise that started ~10 s post handgrip task perception of fatigue does not differ anymore, was really a difference in fatigue state induced between the two experimental conditions?

Another question: could the rating of fatigue associated with the scale instructions be specific to the muscle group involved and therefore not be a general feeling of fatigue when two different tasks are used sequentially? I am not sure the scale used was validated in such an experimental design. Is it possible that the authors highlight a limit of the fatigue scale used in this study?

6. Why did the authors decide to use this specific fatigue scale and not a visual analog scale of fatigue or another tool classically used to investigate fatigue? A better justification of the choice of this scale to investigate fatigue could help the reader.

SPECIFIC COMMENTS

- Abstract line 5: “perceptual (effort)”. Please indicate perception of effort.

- L22-25: I respectfully disagree with this statement. As presented in point 4 above, the literature on the interaction between mental fatigue and physical performance extensively monitored changes in perception of fatigue induced by prolonged mental exertion and its effect on physical endurance.

- L36: the use of the reference Inzlicht et al. 2018 The effort paradox to justify the statement on perceived fatigue is not appropriate as the reference Inzlicht focuses on effort and not fatigue. Please use another reference to avoid confusing the reader between the constructs fatigue and effort.

- L80-83: thank you for this clear justification of the sample size

- L114-116: thank you for this information. It is nice to see such detail being considered in the experimental design, i.e., influence of encouragement on performance.

- L160. “Apparatus and procedures”: please add the information in the main manuscript and not in supplemental material. Such information is crucial for the reader to understand what was done, and the reader should not have to access to a supplemental material to obtain this information when the journal does not impose a word limit. Is a word limit imposed by plos one ?

Brown, D.M.Y., Graham, J.D., Innes, K.I., Harris, S., Flemington, A. & Bray, S.R. (2020) Effects of Prior Cognitive Exertion on Physical Performance: A Systematic Review and Meta-analysis. Sports Med, 50, 497-529.

Temesi, J., Arnal, P.J., Davranche, K., Bonnefoy, R., Levy, P., Verges, S. & Millet, G.Y. (2013) Does central fatigue explain reduced cycling after complete sleep deprivation? Med Sci Sports Exerc, 45, 2243-2253.

6. PLOS authors have the option to publish the peer review history of their article (what does this mean?). If published, this will include your full peer review and any attached files.

Reviewer #1: **Yes: **Robin Souron

Reviewer #2: **Yes: **James Steele

Reviewer #3: No

---

## [Author Response · Author response to Decision Letter 0]

19 Oct 2021

To the Editor and reviewers, 

On behalf of myself and my co-authors I would like to sincerely apologise for the delay in returning this submission to you. 

Please find our responses to your comments below (Line numbers refer to the manuscript with the changes tracked). 

Editor:

We have updated the manuscript to fit PLoS ONE style guides 

This has now been included at the end of the manuscript

Reviewer #1:

This article investigated whether subjective intensity of perceived fatigue, pre-induced through prior upper body activity, differentially impacted performance and altered perceptual (effort) and affective responses during a sustained, isometric contraction in lower body. The authors also explored whether interoceptive awareness moderated the relationship between the perception of fatigue and effort.

I want to congratulate the author for the quality of this very rigorous and well-written scientific paper. This article brings interesting data on how perceived fatigue and effort perception could interact and influence endurance performance. I was naïve on the interoceptive awareness concept, and I think that this strengthen the interest of this paper. I only have some minor comments (and sometimes some personal observations that do not require any answer).

Thank you for your kind words. We would also like to extend our gratitude for your considered and helpful comments in this review. 

Introduction:

Lines 15-16 : perhaps providing a definition of what perception of effort is should strengthen what you say in the first paragraph, i.e. perception of fatigue should be distinguished from perception of effort. I think that in the definition of “perception of fatigue” it should be stated that this state is independent of any previous physical activity.

We have attempted to provide a broad description of the constructs without explicitly stating the definitions within the introduction, because we acknowledge that there are multiple definitions for both the subjective experience of fatigue and effort which would necessitate more detailed description. As suggested, we have now included further description of the distinction between perceived fatigue and effort. 

Lines 9-13: “However, although related, perceived effort should not be used as a synonym for perceived fatigue since they represent distinct constructs [9,10]. For example, an important property of perceived fatigue is that it can be experienced at rest or in the absence of overt activity [11], which may be contrasted to effort, which is experienced in reference to some goal-directed action [9,12–14].”

Line 20: maybe a clear definition of what “self-efficacy” is will make this part clearer for naïve readers

This has now been added

Lines 25-29: “The perception of fatigue has been identified with individuals’ perception of their capacity to effectively exert control through action; that is, ones’ self-efficacy [22]. Self-efficacy defines an individual's task-specific judgment of their capability to execute an action and attain a desired outcome [23]. Accordingly, fatigue and self-efficacy have recently been proposed to be extensionally equivalent [14].”

Line 35: give some example when talking about changes in regulatory responses and performance.

This has now been added

Lines 45-47: “It is possible that incremental increases in perceived fatigue may evoke proportional changes in central regulatory (e.g. perceived effort and affective valence) responses and task performance (e.g. time to task failure).”

Line 54: give example of interoceptive signals

This has now been added

Line 59-62: “Interoception, defined as the encoding and representation of signals (e.g. hormonal, immunological, metabolic, thermal, nociceptive, and visceromotor [34]) reporting the physiological condition of the body [35], is a fundamental component of adaptive (allostatic) behaviour [34].”

Material and methods:

The quality of the figures is very low, especially the one of the figure 1 (i.e. the experimental design). It is impossible to correctly read this figure, please improve the quality

We have updated the figures and checked that they meet with the requirements of PLOS ONE by using the PACE digital diagnosis tool. 

Line 156: it is not clear for me; ROF was assessed 1 min after the beginning of the endurance exercise, and at the end. That’s it? Or was it also recorded every 30sec from the 1st min of the exercise?

RoF was measured at 4 points throughout each session. It was assessed at baseline (i.e. prior to the fatiguing handgrip task); it was assessed at the end of each contraction sequence performed during the handgrip task; during the knee extensor endurance task, RoF was recorded 1 minute into the contraction; It was also assessed upon task failure. 

We chose this particular protocol for several reasons: 1) a limitation of our previous study (Greenhouse-Tucknott, Wrightson, et al., 2020) was that RoF provided at the end of the handgrip task was conducted in a different state (i.e. rest) compared to effort/affective responses recorded during the contraction. This may have influenced rating behaviour and the subsequent relationship between constructs. We attempted to address these potential concerns in this study; 2) we recorded RoF at 1 min because that is the point at which perceptual (effort) and affective responses differed in our previous study; 3) we did not record RoF throughout the endurance task as its dynamics were not central to the main research question and we also believed that providing ratings for three separate scales at the interval set would be too demanding and potentially introduce bias across scales (Raccuglia et al., 2018). 

Maybe change the subtitle “rating of fatigue (RoF)” for “perception of fatigue”. It will be clearer, since the next subsection is called “perception of effort”.

This has now been changed, but the wording of the heading has been changed to “Perceived fatigue” in line with the comment made by reviewer 2 (Line 213).

Results:

Table 1: what does “A.U.contraction-1” means?

This was the average incremental rating response (A.U.: arbitrary units) given per contraction during the handgrip task (described in lines 292-295). We have titled this more clearly in Table 1. 

Line 292-295: “To compare the average rate of change in perceived fatigue between the two experimental manipulations, an estimate of perceived fatigue accumulated per contraction sequence was derived from the difference between the baseline perceptual intensity and the intensity at the end of the HG task, divided by the number of contraction sequences completed.”

Lines 297 and 299: what are Mdn ? Median I guess. Maybe the mean value of the ROF could be helpful to see how far it was from the 6 values for the MOD condition

Yes it reflects the median value. This abbreviation was introduced in the statistical analysis section (Line 380). We have added the mean score for further description of perceptual responses. 

Line 405-411: “In SEV, final ratings (mean=8.0, Mdn=8, range=8-9) were not different from the target value (RoF=8; p>0.999). All participants, except one (RoF=9), had the HG test terminated after reporting an RoF of 8. Conversely, the RoF in MOD (mean=5.4, Mdn=5, range=5-6) did differ from the target value (RoF=5; p=0.001). Eleven participants reported an RoF of 6 at the end of the HG. Despite some exceeding the specified value, the RoF did not exceed the moderate intensity band (>RoF 6; p>0.999). In CON, RoF remained low (mean=1.4, Mdn=1, range=0-3) and did not exceed a RoF of 3 (p>0.999).”

The quality of fig 2 to fig 6 is better than the one of fig 1 but could be much more improved

As above, we have changed the quality of the figures as per the journal’s requirements. 

Paragraph 3.3. It could be a good idea to give the exact time (in sec) for KE endurance performance. We can guess it by checking the fig 2, but it worth it to give the exact values in the text

We agree. This has now been added.

Lines 424-428: “Participants’ ability to sustain the KE endurance task was significantly affected by the experimental manipulation (Fig 2; F(2,58)=11.1, p<0.001, ηp2=0.278), with a shorter performance in MOD (219 ± 68 s, -9.5 ± 19.1%; t(29)=3.11, p=0.013, dav=0.37) and SEV (206 ± 59 s, -13.7 ± 17.2%; t(29)=4.29, p<0.001, dav=0.57) compared to CON (246 ± 79 s). Endurance performance between MOD and SEV was not statistically different (t(29)=1.62, p=0.348, dav=0.20).”

The results section is already long, I am not sure that table 2 is really needed. Maybe it could better fit in the supp data

The changes made to the results section due to points raised by reviewer 2 we believe have reduced the length of the section. However, we understand your point and believe the paper in general is relatively long. We have thus moved the table to the supplementary information (S3 Table). 

Figure 5 should have panel, and you need to use these panels in the text. Further, the legend of one of the panel of the fig 5 is written with a very small police, this is illegible.

Due to recommendations offered by reviewer 2, the analysis changed and subsequently figure 5 has been removed from the manuscript. 

Title of the part 3.6: interoceptive and not interceptive

This has been amended

Discussion:

Lines 439-442: I guess that this sentence could go in the conclusion rather than in the first paragraph of the discussion

This sentence has now moved and added to the beginning on the conclusion (lines: 815-817).

Line 450: while I agree that it was not the aim of this study, it could be good to give some examples for each causes so as to have a general overview of the mechanisms that account for the deleterious effect of prior physical activity on endurance performance in a remote muscle group

We have now included a very brief example to better describe putative causes. Halperin et al. (2015) (and now also Behm et al., 2020) discuss these mechanisms in much greater detail and we hope any interested reader feel compelled to read these articles. 

Lines 612-616: “Several mechanisms accounting for the deleterious effect of prior physical activity on endurance performance in a remote muscle group have been proposed, including neural (e.g. inhibition of descending drive to non-activated muscles), biochemical (e.g. migration of accumulated metabolites) and cognitive factors, though the precise mechanism remains unclear [72].”

Lines 466-473: a personal comment, that is a very interesting paragraph

Thank you. We are glad the manuscript has provided some interest. Due to the changes incurred through the update to the statistical analysis, this section now discussed in relation to the interpretation of the interoception analysis (beginning line 719).

Line 494: “a new task”; you mean the transfer from handgrip exercise to KE exercise?

Yes. We have added this for clarification.

Line 666-669: “According to the authors, changing tasks may attenuate this signal and the need to engage such processes [73], thus a reduction in perceived fatigue coinciding with a new task (i.e. the KE endurance task) in the SEV condition may not be entirely surprising.”

Line 515: which ones?

Due to the changes made to the statistical analysis and the outcome of those analyses, the paragraph which this relates to was removed from the manuscript. 

Lines 520-525: personal comment: another experimental session with the HG exercise until failure (or until a maximal RoF score) could have been interested to meet with what you performed in your previous study (ref 25).

We agree. Unfortunately, we did not have the resources to examine a greater range of the perceived fatigue continuum in this study. It would be interesting to examine whether being forced to stop a task for failing to meet task demands exacerbates the dynamics of fatigue and its relationship to perceived effort and affect (and then performance) during subsequent performance of a new task, whilst controlling for the physiological perturbations incurred (e.g. reductions in HG force generating capacity).

Line 573: is there any possibility for people with “low” individual interoception awareness to improve/train this parameter? Because this interoceptive state can moderate the influence of the perception of fatigue in the perception of effort (thus on the endurance performance), it could be useful to give some recommandations on how a subject with low individual interoception awareness could become a subject with high interoception awareness. I am naïve with this concept, so maybe my comment does not make any sense

This is a very good point, which makes perfect sense, and is a topic of considerable interest within the study of interoception (e.g., Khalsa et al., 2018; Wallman-Jones et al., 2021; Zarza et al., 2019). It is particularly relevant as altered interoceptive characteristics are found in various clinical populations such as anxiety and depression disorders (Avery et al., 2014; Stewart et al., 2014; Zoellner & Craske, 1999), Tourette’s syndrome (Rae et al., 2019), multiple sclerosis (Manjaly et al., 2019), and conditions of body dysmorphia (Ainley & Tsakiris, 2013; Khalsa et al., 2015). Interestingly, there is reasonably consistent evidence to show that transient improvements in interoceptive accuracy can be achieved through increased arousal or attention to self, particularly in individuals with low baseline cardioceptive accuracy (e.g., Ainley et al., 2012; Durlik et al., 2014; Jones & Hollandsworth, 1981; Khalsa et al., 2009). However, to the best of our knowledge, these passing effects have only been examined in the context of interoceptive accuracy with little consideration given to other dimensions of interoception, such as interoceptive awareness. 

Establishing long-term, stable manipulations to interoception appears to be more challenging. Indeed, Oliver Cameron has previously stipulated that interoceptive accuracy may represent an invariant constitutional trait (e.g. like personality) owing to its apparent resistance to experimental manipulation (Cameron, 2001). This viewpoint persists as the accepted position within the literature (e.g., Khalsa et al., 2018; Garfinkel & Critchley, 2013). However, despite this apparent resistance, there is some observational evidence that suggests that physical fitness (and its associated implications for cardiac morphology and function) may be positively associated with cardioceptive accuracy (Jones & Hollandsworth, 1981; Perakakis et al., 2017). This positive association has also been reported for respiratory interoception, suggesting that this effect may be generalisable across different interceptive channels (Faull et al., 2018). However, these effects are not consistently observed (Machado et al., 2019; Montgomery et al., 1984) and the direction of the relationship between aerobic fitness and interoceptive accuracy remains contested (e.g., interoception may influence physical activity behaviour which has implications for aerobic fitness; see Georgiou et al., 2015). Moving beyond observational studies, experimental evidence for this effect remains scant. Indeed, in our own lab we have observed null effects using a short-term exercise training intervention on dimensions of cardiac interoception (unpublished). 

Another avenue that has received attention is the use of mindfulness and meditation to induce stable changes in interoceptive accuracy. While there are some examples to show that mindfulness can improve interoceptive accuracy (Bornemann & Singer, 2017; Farb et al., 2013; Meyerholz et al, 2019) the use of mediation appears to be ineffective; instead, meditation may augment a person’s perceived interoceptive sensibility (Khalsa et al., 2020). Changes in interoceptive sensibility have also been reported following a 12 week exercise intervention (Mehling et al., 2018). It is not clear whether mindfulness, mediation, or physical activity influence stable changes in interoceptive awareness. 

Given careful consideration of the present state of the evidence in this area, as briefly outlined above, we currently conform to the conventional position that interoception represents a relatively stable component of individual emotional reactivity. Further, we choose not to include this argument into our manuscript as 1) we do not believe that it is particularly relevant to the research question, and 2) the need to provide a sufficient appraisal of the points raised above would make the text excessively long and detract from the main points of the study. 

Reviewer #2:

Firstly, my apologies for my tardiness in returning my review! Secondly, I just want to say I am continually impressed by the great work being completed by your research group and I’m honoured to be afforded the opportunity to review and provide feedback on it. Below I’ve made notes from my reading of the manuscript. Most are fairly minor to be honest or just annoying quibbles about terminology which are probably to be expected from me. Others are just some of my thoughts and not necessarily in need of being addressed in this manuscript specifically. Lastly, some comments pertain to the statistical analysis and presentation of data. Given this is an open peer review I would be happy to arrange to call and discuss any points highlighted and to offer assistance with any of the analyses should you feel they are worth running. If not then no problem, however I make a note also regarding the data availability and addressing this would mean that readers can look to run some of the suggested analyses (or others) if they wish (and I might at some point).

Thanks,

James Steele

Thank you so much for your kind words. Your comments and recommendations have helped us to develop our own understanding particularly of the adopted statistical methods, and have helped to significantly improve the manuscript. We would be keen to continue to discuss this work and topic in general in future. 

Comments:

Minor quibble, and to be honest I'm not certain it's really that worth changing throughout but I'll mention it anyway... strictly speaking 'perception of' as opposed to 'perceived' is probably more appropriate. Olivier Massin noted the following in feedback on an early draft of my conceptual definitions paper:

"...is subjective effort the perception of effort of perceived effort? You say both, but they can’t be the same (the perception of a tree is not the perceived tree). I would suggest you go for the latter, as I’m not sure whether there are many cases in which the perception of an x of kind K is itself of kind K (and I’m quite sure no effort is a perception, while I fully agree that some effort are perceived or felt)"

As I said, a minor quibble and one probably of bigger concern to analytic philosophy than necessarily in empirical work such as this where people know what your referring to. So no real need to address... but I just find it interesting to make people aware.

Thank you for highlighting this, though we are slightly confused by your stance. You suggest that “perception of” is probably more appropriate than “perceived”, but the response from Massin appears to suggest the opposite? We have changed the terminology to “perceived effort/fatigue” throughout the manuscript (where deemed appropriate) as that would appear to refer best to the experience of the participants. 

Line 16: Sensations of tiredness… I'm not sure I would label these as 'sensations' as it is not clear what stimuli are being sensed, and further that sensation isn't phenomenological, but perceptions, feelings are. Maybe the latter is a better label here.

Agreed. We have now changed this sentence. 

 Line 21-23: “Perceived fatigue is conventionally associated with feelings of tiredness, a lack of energy, exhaustion and a desire to rest [18,19].”

Line 19: I'm not sure I'd say it's a phenomenological counterpart given that self-efficacy is also phenomenological... this is why I've posed that perception of fatigue and self-efficacy are extensionally equivalent and merely opposite signed transformations of one another. Also, I would perhaps define self-efficacy here considering that it is in different places used by people as a state or trait construct, and also that some people consider a 'general' self-efficacy, whereas Bandura's original work for example defined it more in relation to the specific capacity to perform a specific task i.e. self-efficacy for X.

As suggested by reviewer 1 as well, the definition of self-efficacy proposed by Bandura (1997) has now been incorporated within the introduction. We have also changed the wording in the description of the association between fatigue and self-efficacy to address your first point.

Lines 25-29: “The perception of fatigue has been identified with individuals’ perception of their capacity to effectively exert control through action; that is, ones’ self-efficacy [22]. Self-efficacy defines an individual's task-specific judgment of their capability to execute an action and attain a desired outcome [23]. Accordingly, fatigue and self-efficacy have recently been proposed to be extensionally equivalent [14].”

Line 32: Ordinal... although It's also not clear to me whether ontologically classifying fatigue into discrete ordinal bins makes sense anyway.

We understand the point made here and share your feelings regarding the classification of fatigue into ordinal bins. Our initial proposition was that the consequences of entering a perceived state of fatigue (irrespective of the actual numerical score used to identify the intensity of that perception on some psychometric instrument) reflect a binary variable. That is, one is either “fatigued” or not after exceeding some threshold. This is similar to a yes/no question. Based on this it was felt that describing fatigue as a categorical variable expressed this best (as similarly done in relation to the effort-value relationship: Inzlicht et al., 2018). We acknowledge that this is not perfect. 

Line 43: Sense… As above, interoception might be better called a perception... similarly to exteroception. Both are perceptions that are resultant from sensation of some stimuli.

This has now been amended.

Line 59-62: “Interoception, defined as the encoding and representation of signals (e.g. hormonal, immunological, metabolic, thermal, nociceptive, and visceromotor [34]) reporting the physiological condition of the body [35], is a fundamental component of adaptive (allostatic) behaviour [34].”

Interoception: I must confess I have not worked with interoceptive awareness measures, but I am not sure I catch the intuition that interoceptive accuracy in one domain (e.g. heartbeats) might be reflective of a general interoception construct. I am aware there are various multi-dimensional measures available - this one for example, though self-report, seems to be of a more general interoception construct https://www.ncbi.nlm.nih.gov/pmc/articles/PMC6279042/

I guess you could argue that, in the vein of my arguments regarding actual-perception of effort, any psychophysical examination of relations between objective and subjective components of some interoceptive construct that do not fall on the identity could be evidence of inaccuracy in interoceptive awareness. Confidence and awareness are sort of akin to that of the Dunning Kruger effect (nice computational model example here - http://haines-lab.com/post/2021-01-10-modeling-classic-effects-dunning-kruger/).

I guess in a round about way I'm just getting at the point of being quite cautious regarding conclusions about the role of 'interoception' per se in your paper as opposed to the relationship between interoceptive ability regarding perception of heart rate and its relationship to your DVs... a slightly more specific conclusion.

We understand these concerns. To the best of our knowledge, few studies have directly investigated correspondence of interoceptive ability across physiological axes. Of the available evidence, good correspondence has been reported across cardiac and gastric interoceptive axes (Herbert et al., 2012; Whitehead & Drescher, 1980), which likely reflects their shared sympatho-vagal afferent pathways and overlapping cortical representation within the insula (Avery et al., 2015; Harrison et al., 2010; Malliani et al., 1986). Less clear is the relationship between cardiac and respiratory axes (Harver et al., 1993; Pennebaker et al. 1985 cited in Vaitl, 1996). However, Garfinkel and colleagues again have provided some evidence that interoceptive awareness (i.e., metacognition) may be transferable across physiological domains. In their study, though accuracy on a heartbeat (heartbeat discrimination task) and respiratory (breathing resistance task) interoceptive task were not related, derived metacognitive awareness across sensory axes were positively associated (Garfinkel et al., 2015). They suggest that knowing whether or not you are “good” or “bad” at judging internal bodily sensations appears to be a relatively stable trait that spans across different interoceptive axes. We have interpreted our results with this finding in mind and have detailed this point in the discussion to aid the reader in understanding the basis of this:

Line 739-740: “The measure of interoceptive awareness used in the present study is cited as a stable representation of ‘error awareness’, that may be generalisable across interoceptive axes [83].”

With respect to the use of multidimensional interoceptive self-report instruments (e.g., Mehling’s MAIA or Porge’s BPQ), we would argue that, although useful, these measures of interoception are limited to the appraisal of interoceptive sensibility (i.e., a person’s subjective trait belief of their interoceptive experience), which may not reflect other interoceptive dimensions. Data reported by Garfinkel et al. (2015) showed that BPQ is unrelated to both cardiac interoceptive accuracy and awareness and is also unrelated to a state confidence measure of interoception (obtained by VAS). Further, to give consideration to some of the literature around mindfulness and meditation, previous studies have shown that these practices may improve markers interoceptive sensibility (Daubenmier et al., 2013) but are less effective when trying to influence interoceptive accuracy or awareness (Khalsa et al., 2020). As a consequence, these multidimensional self-report tools can be considered to measure a distinct aspect of interoceptive experience that is independent of the measures reported in the present study. 

We believe that we have been sensitive enough in describing the effects to outline a broad conceptualisation of the role of interoception in shaping (or not) the subjective experiences reported during physical activity. 

Lines 85-87: Did you conduct a debrief with participants and check whether or not they had discerned the purpose of the study or if the deception was maintained?

No we did not conduct a formal debrief to assess this. Informally, the participants appeared naive to the specific aims of the study. Some did identify that they were aware of relationship between the duration that the handgrip task was performed for and their rating of fatigue, but again were unaware of the specific aims of the study. 

Line 102: Can you note the duration (seconds) of the MVCs?

 This has now been added.

Lines 127-129: “Participants were then asked to perform a series of brief (5s) maximal voluntary contractions (MVCs), alternating between hands (one minute separating each contraction).”

Line 118: It’s not clear how it was blinded... participants were deceived as to the overall aims as noted, but surely participants knew which condition they had experienced each time they came in? Or do you mean to say that they didn’t know until they stopped the HG task at a given RoF? In which case they would have been unblinded during the KE task.

Yes, that is correct. On arrival they only knew that they would be performing the handgrip for an unknown period of time (that is until they were told to stop; they did not know the stopping criteria). They would have known the condition just performed in the following endurance task. We have removed the description of single blind from the materials and methods section (line 145). 

Jamovi: Can you add the jamovi file/code to the supplementary materials along with the data in the format used for analysis so that it can be easily reproduced?

We have now uploaded all files which include the data used. 

Data: So the online materials in excel format are not particularly easy to navigate. Also, I don’t think all the raw data are there. If possible I would re-organise the data (following for example the guidance here - https://www.tandfonline.com/doi/full/10.1080/00031305.2017.1375989). If you had it in a specific format for analysis then I would include that as a separate file and indicate this clearly.

Thank you for highlighting this paper. We have changed the file so the layout is consistent (wide format) throughout and have renamed the tabs to clearly indicate the variable of presented. We are surprised that you think that some raw data may be missing? We hope that this is now rectified. 

Line 248: It's not clear why it was necessary to use a mixed effects model here. You note that it is flexible to imbalances and missing data, but as you removed data to ensure it was balanced across participants. Though, I do wonder whether you might benefit from the inclusion of random slopes to your model for time... individuals may differ in the growth of effort/affect over time because of different rates of fatigue for example for the former.

We chose a mixed effects model here because: 1) due to individual differences in performance time, this resulted in a different number of ratings/participants across each time point (particularly the later time points) between conditions. Mixed effects model enabled us to model this without losing excessive data as would be the case with a repeated measures ANOVA; 2) the ability to model the random variance associated with the individual participants. However, though the mixed model enables us to effectively deal with missing data without losing power, contrasts (or post hoc comparisons) could not be performed on the full data set above time points exceeding 300 s. This was simply because there was no data in particular conditions (e.g. SEV) above that time. Since these points held little relevance to the research question, the data was reduced to that containing data in which comparisons between conditions could be made. Within that, not all participants completed the same duration (some <300s), so the mixed effect model still served to account for missing values across certain individuals (that is, the data wasn’t balanced even with this removal of data). 

In relation to your second point, using a data driven approach, we compared two models: one including just the random intercept across participants and the other including both the intercept and slope of time as random effects. For the modelling of effort ratings it was shown that the intercept only model was a better model than the model that included both random intercepts and random slopes for time across participants (Intercept: AIC – 1739.700; Intercept + Time Slope: 1774.698). For affect, the opposite was true (Intercept: AIC – 2020.738; Intercept + Time Slope: 1910.484). We have updated the methods to demonstrate this consideration and altered the results/discussion accordingly. 

medmod: Can you clarify... was this moderation or mediation that was explored? If just moderation then you could have just explored this with a general linear model in base jamovi. Or preferably a mixed model.

Yes, we used the medmod module to perform a moderation analysis. It returns the same output as a general linear model (except you have to centre the data manually). We did not consider using a mixed effect model. But based on your recommendation (below) we re-ran the analysis using a mixed model. 

Figures: Given the within participant design, it would be better to present the figures with individual low alpha/transparency lines for each participant across the predictor variables in each figure. I would also present interval estimates i.e. CIs. Actually, I would add this throughout the results.

We have now updated the figure for perceived effort and added one to for affective valence across the endurance task. We could only present low alpha/transparency lines for each individual using smoothed line plot. 

Figure 2: I can’t see the data for this for example in the online file.

This is odd. It is within the “TTF” tab on the online data spreadsheet. 

Section 3.4: Interesting, and surprising how quickly the SEV RoF dropped and equalised between conditions (I wonder if this occurred within ~10 seconds between HG and KE, or it occurred during the first min of the KE task). Although, Whittaker et al show drops of say ~1-3 pts on their RoF scale during rest periods (see their figure 6; https://www.sciencedirect.com/science/article/pii/S1050641119300136?via%3Dihub). I guess this might explain the lack of difference in performance and perception of effort during the KE task between MOD and SEV as you note.

Unfortunately we cannot determine the dynamics of RoF from the end of the handgrip task to the rating provided during the endurance task (i.e 1 minute). We think it is likely that a large portion of the sudden drop in perceived fatigue comes from experiencing the new task and the effect of experiencing a task to feel “easier” than expected or the fact that the new task leads participants to feel that they can exert control more efficiently than expected. The declines during rest shown by Whittaker et al. are surprising, however. Establishing the kinetics of the perception of fatigue is an area of potential to be explored, but (as stated) was beyond the scope of the present study.

Figure 4: I find it quite interesting that perception of effort often follows a linear, or convex function... assuming my definition of actual effort, and linear fatigue (which I think is often the case in tasks like this, at least we've found this to be the case in the lab, or in the model simulated in my pre-print [figure 4(b)] where it is proportional to effort at t-1), actual effort increases with time in a concave function. Interestingly though in a few models I've been exploring to examine effort psychophysics on the same ratio scales for actual and perception, isometric endurance tasks display this convex function over time, yet another model of dynamic tasks (elbow flexion with a dumbell) has shown a function a bit closer to the concave actual effort function. I suspect this is something to do with the scale and perhaps ceiling effects, and perhaps also that isometric tasks are inherently more uncomfortable to perform. Of course, you differentiated effort and discomfort here, as do we in our work. But although we have found in several studies that lighter loads/torques to task failure in resistance type exercise cause greater discomfort compared to heavier in dynamic tasks, the data we have for isometric tasks suggests they are similarly uncomfortable (e.g. https://pubmed.ncbi.nlm.nih.gov/32461833/ and also some data for another study we are writing up). It makes me wonder if, even with instruction, people sometimes still 'taint' their ratings of perception of effort with the discomfort of the task when it is particularly salient such as in isometric tasks. Anyway, just some thoughts to share here... not necessarily anything to address given the aims.

This is interesting. We agree about concerns surrounding ceiling effects with present scales. We used the CR10 scale for the measurement of perceived effort. Though participants had the option of rating effort greater than the maximal numeric value (therefore perceiving effort greater than previously experienced) it is our experience that this is rarely chosen. 

R.E. effort and discomfort. We think your point makes a lot of sense. Though as you highlight participants may be aware of the distinction between effort and discomfort, in many scenarios it is possible that the latter maybe used as a heuristic in relation to “effort” (that is, this is uncomfortable thus it must require effort) or functions at least to calibrate it. As you say, in uncomfortable situations it is possible that like fatigue, discomfort is some weighting factor than influences subjective effort rating behaviour. Too often systems are separated (e.g. proprioceptive and interoceptive) in relation to the perceptual responses arising from them. We believe that there is likely to be substantial interactions between different systems that provide the ‘colour ‘of our perceptual and affective experiences. 

Affect: Similarly, to figure 4, one for affect over time would be of interest to show. Or at least have this data available.

This has now been added to the manuscript (Fig 5). 

Section 3.5: Have you considered comparing these using something like cocor (http://comparingcorrelations.org/)? Although, It's not clear why you would worry about running these separately by condition. You could employ a mixed model approach for repeated measures correlations and calculate this in a single model e.g. https://www.frontiersin.org/articles/10.3389/fpsyg.2017.00456/full. If you felt that were effects specifically by condition categorically you could use similar methods to those used by Vigotsky et al here (code available in their supplementary materials - https://peerj.com/articles/5071/#supp-3) to employ a mixed effect model including condition as an effect. Though I think you should probably just keep things continuous and just color code the figure to show condition.

Thank you for your advice. We had considered this initially. We have now adopted a mixed approach using repeated measures correlations:

Lines 510-515: “Correlations were evident between time to task failure and perceived effort after the first minute (rrm= −0.46 [95% CI: −0.64 to −0.23], p<0.001), the rate of change in effort (rrm=−0.41 [95% CI: −0.60 to −0.17], p=0.005), the rate of change in affective valence (rrm=0.38 [95% CI: 0.13 to 0.58], p=0.014) and perceived fatigue after the first minute (rrm=−0.54 [95% CI: −0.70 to −0.32], p<0.001). Time to task failure was not related to affective valence reported after the first minute across conditions (rrm=0.20 [95% CI: −0.06 to 0.44], p=0.166).”

Figure 5. I would clarify this as being initial ratings of perception of effort in that panel.

As a result of the recommendations above, this figure was removed from the manuscript.

Table 2: Presumably, this is a grand mean intercept for these models? If this is what jamovi does as standard it might be worth clarifying this is the case in the methods.

We have added this to the materials and method section.

Lines 333-335: “The modelling of random effects was initially compared across two models: one in which random intercepts (grand mean) were included across participants and one in which random intercepts and random slopes for the effect of time varied across participants.”

Section 3.6: As noted, I wonder if a better approach here instead of averaging across MOD and SEV would be to model it all together, along with CON. You could just simply explore it in a model like: effort ~ RoF + Int_awareness + RoF*Int_awareness + (1|Participant). You could also add back in the Condition if you really think that interoceptions moderating effect would be moderated itself by this. I can't think why it would be though.

We have re-run the analysis as recommended and updated the statistical analysis and results/discussion as a consequence. 

Lines 368-375: “Finally, exploratory analyses were performed to assess the influence of interoception on perceptual and affective constructs. First, LMM were used to examine whether interoceptive dimensions predicted RoF, perceived effort and affective valence during the initial stages (i.e. at 1 minute) of the endurance task across conditions, with intercepts entered as random effects across participants. Next, the moderation effects of dimensions of interoception were examined on the ability of RoF to predict perceived effort/affective valence. This was performed again using a LMM with the intercept entered as a random effect across participants.”

 Lines 538-554: “We explored the potential influence of interoception on the perceptual and affective responses to the endurance task and its potential moderating effect on the relationships between constructs. First, we examined whether interoception predicted perceptual and affective responses recorded during the endurance task. Though interoceptive accuracy (estimate = 1.57, [95% CI: −1.66 to 4.81], p=0.348) and confidence (estimate = -0.02, [95% CI: −0.34 to 0.31], p=0.920) did not, interoceptive awareness was shown to predict RoF reported during the endurance tasks (estimate = -4.18, [95% CI: −8.08 to -0.28], p=0.048). That is, individuals with greater awareness of resting heartbeats reported lower RoF during the endurance task. None of the dimensions of interoception predicted perceived effort or affective valence recorded at the same time point during the task (Tables 2 and 3). 

We speculated that individuals’ meta-awareness of interoceptive signals may be particularly important to the relationship between perceived fatigue and effort/affective valence. There was no significant interaction effect between RoF and interoceptive awareness on the prediction of perceived effort (Table 2). This was the same when both interoceptive accuracy and interoceptive confidence were used as the moderator variable (Table 2). Moreover, the same effects also extended to the prediction of affective valence (Table 3).“

Lines 412-419: This is also interesting, and the opposite of what I would have expected to be honest given I'd expect those with greater interoceptive awareness to have more accurate perception of fatigue, and thus more accurate perception of effort.

Due to the changes made to the statistical analysis, the moderating effect of interoceptive awareness on the ability of RoF to predict perceived effort was lost. However, we observed that interoceptive awareness did display a negative relationship with RoF. In keeping with your point, this may appear surprising as it may be intuitive to think that greater interoceptive awareness will lead to a more accurate and perhaps more sensitive perception of fatigue. We (Greenhouse-Tucknott, Butterworth, et al., 2020) have recently drawn on conceptualisations of fatigue as a feeling arising from metacognition, relating to the detection of a consistent mismatch between the brain’s predictions concerning internal state and the feedback received concerning the actual state of the body (Stephan et al., 2016). Greater awareness of internal representations of the body may mean that predictions are given more “weight” or confidence, such that they suppress ascending feedback and thus limiting the effect of error in generating a state of perceived fatigue. Accordingly, greater awareness or interoceptive states may attenuate the development of fatigue. We have outlined this in more detail in the discussion:

Lines 723-749: “Drawing upon recent descriptions of chronic, pathological fatigue [22], we have previously proposed that the subjective symptom of fatigue arising from acute physical exertion may do so as a result of continued detection of challenges that undermines the experience of control over the body [79]. These challenges are underpinned by discrepancies between top-down expectations or predictions of internal states and the sensory evidence received from the body. Reduced confidence in held predictions may subsequently result in greater disparity between what the brain predicts the physiological condition of the body to be and its true state, which subsequently alters the processing of error signals, resulting in greater perceived effort [80,81] and increasingly negative affective states [82]. Under this framework, we provide a theoretical account of not only how the perception of fatigue emerges during acute physical activity, but also why subsequent activity may be perceived to be more effortful and less pleasurable than normal. The subjective experience of fatigue is offered as an experience emerging from higher-order, metacognitive processing [22] and closely aligned to changes in the estimated precision of descending efferent predictions [79]. In line with this proposition, interoceptive awareness (a measure of an individual’s meta-awareness of interoceptive signals) predicted RoF during the endurance task. Specifically, the greater one’s interoceptive awareness, the lower the subjective experience of fatigue reported during the endurance task across all three trials. The measure of interoceptive awareness used in the present study is cited as a stable representation of ‘error awareness’, that may be generalisable across interoceptive axes [83]. The results indicate that the greater the awareness of internal body representations, which one may assume suggests greater confidence in interoceptive predictions of internal states, the smaller or less reliable prediction error is believed to be, resulting in an attenuated experience of fatigue. Importantly, this was evident for perceived fatigue only, with neither effort nor affective valence related to any dimension of interoception. This conforms with previous studies examining cardiac interoception on perceived effort [84] and affective valence [85] during physical activity. Indeed, though we were unable to disassociate constructs based on their relationship to performance, this finding indicates that the studied constructs may involve different circuitry, with metacognition playing an important role specifically in the experience of fatigue.”

Lack of statistically significant differences: I wonder if it is perhaps worth considering looking at whether there was in fact statistical equivalence between MOD and SEV for rating of perception of effort and performance. You would of course need to determine a justifiable smallest effect size of interest for the difference in performance and effort between the two. This is a bit tough given it will obviously be post hoc. But you could base this on reliability of the constructs. You could do the same for the RoF, though it’s not clear what the reliability is to my knowledge… though you could make a reasonable assumption that it’s probably similar to RPE given the similar construction of the scales in terms of points.

Given the RoF response during the endurance task and the difficulty in interpreting the (perceived) fatigue-performance relationship, we decided against this, since it would not help decipher this further. What is required is to test this hypothesis again but ensuring fatigue is differentiated between tasks throughout. 

Lines 516-519: Might be worth noting changes in affect with time during other multi-joint tasks to TF e.g. https://journals.humankinetics.com/view/journals/ijspp/16/1/article-p135.xml

Thank you for bringing this work to our attention. Due to the changes in statistical approach this paragraph of the discussion was removed. 

Reviewer #3:

In this study, the authors aimed at investigating the effect of various intensities of perception of fatigue on subsequent endurance performance. To do so, they performed a prior handgrip task until the participants rated two different intensities of perception of fatigue (moderate vs severe) via a recently validated rating of fatigue scale. Then participants had to perform a submaximal isometric contraction at 20% of their maximal voluntary contraction peak torque. The authors observed a reduced endurance performance following prior physical exertion, with no difference between the moderate and severe fatigue conditions. They concluded that perception of fatigue indirectly limits endurance performance by making a physical task appearing more effortful.

I commend the authors for performing a study focusing on the influence of the subjective aspect of fatigue on subsequent endurance performance. This approach is from my point of view important as fatigue manifests both objectively and subjectively, and to the best of my knowledge, the literature focuses mainly on the objective manifestation of fatigue.

Thank you for your acknowledgement of the interest of our work. 

Please find below some comments/questions.

1. The authors used prior physical exertion to increase the perception of fatigue. However, physical exertion does not only induce an increase in perception of fatigue but also induces a decrease in force production capacity of the working muscles. Therefore, both subjective (perception of fatigue) and objective (decrease in MVC) aspects of fatigue were induced by the experimental protocol. How did the authors control for this confound?

This is an important point; one that has informed our rationale across a series of studies we have recently conducted. This study was conducted in light of our previous findings utilising the same experimental protocol (Greenhouse-Tucknott et al., 2020). In our previous study, we conducted two experiments examining the functional implications of performing a motor/physical task in one part of the body (i.e. the upper body using handgrip exercise) on another, remote part (i.e. the knee extensors). In the first experiment we showed that a demanding prior handgrip task reduced participants’ capacity to sustain a submaximal contraction of the dominant knee extensors. The prior task served to increase perceived fatigue immediately before the start of the subsequent (knee extensor) endurance task, which was associated with heightened perceived effort and a more negative affective valence during the protracted task. The altered perceptual and affective response to the task was shown to be associated with individuals reduced ability to tolerate the task. In the second experiment, we assessed whether the prior handgrip task impacted the capacity of the neuromuscular system to generate force, as this may have influenced the participants perception of the task and made the comparison between condition not like-for-like. Using transcranial magnetic stimulation, we found little evidence that the prior task impacted the neuromuscular function of the dominant knee extensors (though it should be pointed out that, tests of equivalence did not support responses between conditions to be exact the same). We therefore cautiously proposed that the prior handgrip exercise elevates the (global) perception of fatigue and impairs subsequent endurance performance in the knee extensors independent of acute impairment of the neuromuscular function of the knee extensors. It is important to point out that this is in line with several other studies (e.g. Aboodarda et al., 2020; Amann et al., 2013; Johnson et al., 2015).

So while it is the case that sufficiently demanding physical tasks do induce a deficit force production of the active muscles (here we show this in maximal force production of the hand during the grip task), the protocol adopted does not induce such deficits in the knee extensors. A recent meta-analysis on the topic of transferable fatigability across different non-active muscles of the body supports our previous assertion (Behm et al., 2021) - there is little evidence of acute deficits in momentary force or “power” incurred in non-local muscles following prior exertion. However, we do note that there was some evidence of differences in EMG response between conditions which may suggest changes in motor unit activation in the KE by the prior task (Table S4).However, as outlined in the discussion of the limitations of our study, we believe that the substantial variation between participants and the limited explanatory power of the model limits conclusive interpretation. Indeed, this result goes against previous evidence utilising similar experimental protocols (Greenhouse-Tucknott, Wrightson, et al., 2020; Amann et al., 2013; Johnson et al., 2015; Morgan et al., 2019). We believe that participants in the present study (most likely) performed the knee extensor endurance task in all conditions with the integrity of the neuromuscular system fully intact.

We have made small adjustments to the introduction to emphasise this point:

Lines 90-94: “We have previously demonstrated that this paradigm enables the effects of a perceived state of fatigue on the performance of a subsequent physical endurance task to be evaluated within an intact system; that is, independent of concurrent neuromuscular deficits typically incurred through protracted physical exertion [29]. This is in line with previous findings [44–46].”

Also added to the limitations:

Lines 801-809: “Furthermore, at odds with our previous study [29] and others [45,46,88], there appeared to be some evidence of an effect of condition on the EMG response in the KE muscles (S4 Table). This may suggest that motor unit activation may have differed between conditions, meaning that the endurance tasks were not performed in equivalent physiological conditions. This may have contributed to altered perception beyond the proposed top-down (meta)cognitive influences [89]. However, it is acknowledged that inferences drawn on central drive from bipolar surface EMG is problematic [90], and we observed large variation between individuals and only moderate explanation of the variance from the model. As such, we believe that interpretation of the present data is difficult and this effect needs to be investigated further.”

2. In line with the previous comments, one important limitation of the protocol is the lack of knee extensor maximal voluntary contraction measurement at the onset of the knee extensor endurance task. Without this measurement, the reader cannot be secured that the prior physical exertion did not reduce the maximal force production capacity of the knee extensors. Such a decrease in maximal force production capacity could alter endurance performance independently of any changes in perception of fatigue.

As highlight in our previous response, we have already demonstrated that the force generating capacity of the knee extensors is not influenced by the adopted handgrip task (Greenhouse-Tucknott et al., 2020). It is important to note that in our original study the handgrip was performed to task failure (i.e. an inability to produce the required force), resulting in task being performed for longer than both conditions in the presented manuscript (mean: ~11 vs. ~5 and ~8 repetitions reported here). This plus the work of others (Aboodarda et al., 2020; Amann et al., 2013; Johnson et al., 2015) highlighted above gives us confidence in the assumption that declines in the force generating capacity of the knee extensors was not a significant limitation of the study. We do acknowledge that this effect was not replicated in the present study and is therefore an assumption, but a valid one given the current evidence. 

3. In line with points 1 and 2 previously mentioned, why did the authors decided to use prior physical exertion and no other methodologies known to increase the perception of fatigue without decreasing force production capacity such as sleep deprivation (Temesi et al., 2013) or mental fatigue/prior prolonged mental exertion (Brown et al., 2020)?

Contrary to prior physical exertion, prior mental exertion or sleep deprivation does not alter maximal force production capacity, and this confound could have been better controlled. The manuscript could benefit from a better justification of the choice of the experimental manipulation, as well as a clear discussion on the limits of the approach used.

As highlighted above, we have shown that the adopted experimental approach does in fact increase the subjective experience of fatigue and influence subsequent performance independent of changes in neuromuscular function within the examined muscle (Greenhouse-Tucknott et al., 2020). Therefore, we do not believe the adopted design represents a confound in our study. We have cited this evidence in the introduction to emphasise this to the reader (see above).

Other experimental approaches (such as those you have highlighted) were considered. As described, sleep deprivation has been shown to increase the perceived effort (RPE) and impair endurance performance independent of functional changes to the activated muscle group’s capacity to produce force (Temesi et al., 2013). The issue we believe with sleep deprivation and the study of the perception of fatigue, is that it requires clear separation of the perception of sleepiness from fatigue, which previously has not been done. The two are often used interchangeably in some contexts, and this appears to primarily be driven by common terms used to describe the two states (i.e. the feeling of being ‘tired’; Shen et al., 2006). However, sleepiness and fatigue are distinct states. This may be evidenced, for example, by their disparate responses to acute physical exertion (that is, while we understand that exercise may exacerbate perceptions of fatigue, it may in fact alleviate perceptions of sleepiness; Leproult et al., 1997; Matsumoto et al., 2002). Here, we demonstrate this distinction by showing participants’ perception of fatigue to be increased following the performance of the handgrip task, while the subjective experience of sleepiness remained low and relatively stable across each trial (of note, we clearly separated the two constructs during the description of the scales by removing all references to sleep or feelings upon waking used in the original instructions developed for the RoF scale; Lines 217-221). In the referenced study, Temesi et al (2013) did not measure the subjective perception of fatigue (only sleepiness). It would appear that elevated perceptions of fatigue and sleepiness may evoke similar perceptual changes (i.e. elevate effort) and impair performance in subsequent physical tasks. However, the important point is that sleep deprivation effects may not be related to the perception of fatigue exclusively, which was the primary focus of the manuscript. 

In regards to prior cognitive exertion and the development of “mental/cognitive fatigue”, the meta-analysis by Brown and colleagues (2020) concluded that current evidence supports a small-to-moderate effect of prior cognitive activity on indices of physical performance, with the largest effects seen when the physical task was force/resistance-based. However, there have also been calls that selective reporting bias may inflate current estimates of the effect of prior cognitive exertion on subsequent physical performance (Holgado, Sanabria, et al., 2020), thus further work is required to establish this effect (we do acknowledge the different inclusion criteria between the work of Brown et al., (2020) and Holgado et al., (2020) which may hinder full comparison between findings). As such, there is need for replication of previously observed effects (e.g. recent work has failed to reproduce some of the seminal findings of the literature; Holgado, Troya, et al., (2020)). We recently examined the impact on prior cognitive exertion on participants’ ability to sustain an isometric, submaximal contraction of the knee extensors (Greenhouse-Tucknott et al., 2021) - the same performance task used in the presented manuscript - providing a conceptual replication of a widely cited effect (Pageaux et al., 2013). This was not a full replication as the adopted cognitive task differed between studies due to our study being part of a wider data collection programme which included multiple research hypotheses. We demonstrated that protracted cognitive exertion served to increase the subjective perception of fatigue, but we did not replicate previous findings, with no effect on performance or perceived effort (or affective valence) evident during the subsequent physical endurance task (Greenhouse-Tucknott et al., 2021). Our findings add to growing uncertainty concerning the effect of prior cognitive activity on subsequent physical performance. 

4. The literature on the interaction between mental fatigue and physical performance extensively monitored changes in perception of fatigue induced by prolonged mental exertion (e.g., see work of Marcora, Roelands, Smith, Pageaux). Surprisingly, the authors did not consider this literature in their manuscript as the mental fatigue literature manipulated perception of fatigue and associated changes in performance.

I think it is important to stress at this point that the aims of the study were not concerned with whether “mental” or “physical” fatigue influenced subsequent behaviour. It was whether the subjective intensity of a perceived state of fatigue evoked distinct responses (perceptual, affective and behavioural) during a physical endurance task. We chose to manipulate perceived fatigue by using a physical handgrip protocol. 

We have highlighted the work of Benoit et al (2019) and Harris & Bray (2019), both of which used demanding cognitive tasks to induce a (perceived) state of fatigue and then evaluated subsequent perceived effort during secondary cognitive and physical tasks (references 30 and 31; Lines 34-37). In both studies (like our original study) they found significant relationships between the intensity of the perceived fatigue induced and the subjective experience of effort reported in a subsequent task. To the best of our knowledge other literature demonstrating an elevation in perceived fatigue following cognitive activity and a subsequent increase in perceived effort during a physical task have not explicitly examined the relationship between the two constructs. These studies assume an association as the task induces a change in perceived fatigue and effort, but this cannot be assumed. This is the reason these references were chosen in relation to the point we were making. 

5. The results indicate that feelings of fatigue did not differ between the moderate and severe fatigue conditions early in the knee extensor endurance task (see figure 3). I understand that the difference was evident at the end of the handgrip task as it was the end task criteria.

Based on this observation, can we really consider that the experimental manipulation was successful? In other words, if early in the subsequent exercise that started ~10 s post handgrip task perception of fatigue does not differ anymore, was really a difference in fatigue state induced between the two experimental conditions?

Another question: could the rating of fatigue associated with the scale instructions be specific to the muscle group involved and therefore not be a general feeling of fatigue when two different tasks are used sequentially? I am not sure the scale used was validated in such an experimental design. Is it possible that the authors highlight a limit of the fatigue scale used in this study?

First it should be pointed out that the change in perceived fatigue in the severe condition occurred over the first minute of the endurance task (so not immediately). As highlighted by reviewer 2 and the work they cite, this is not an uncommon finding after the removal of a particular stressor. The notion that fatigue (though mainly in relation to its detrimental performance effects rather than its perception it must be said) can change with the engagement of a new task is also not novel - David Hockey discusses evidence of this in his excellent book (Hockey, 2013). He claims that there appears to be two separate consequences of continuous work: the need for a change of task goal (not necessarily rest) and the need to reduce executive activity (Hockey, 2013). The change of task in the present study may have satisfied the first need. 

We do believe a difference in (perceived and physical) state was induced between the two experimental conditions. In addition to the perceptual ratings, maximal force-generating capacity in the handgrip task was also significantly reduced in the severe fatigue condition indicative of the greater stress placed upon the participants. We have acknowledged in the discussion of the limitations of the study that we have not assessed the full fatigue-performance relationship (lines 797-800). Further research is required to examine this effect against additional markers of physiological stress and arousal across the whole range of perceptual intensities to see how these responses change with the initiation of a new task. What we believe we have demonstrated is an often neglected and understudied (particularly in the exercise sciences) response related to perceived fatigue, which may be due, at least in part, to the use of multi-item instruments to capture perceived fatigue, the lack of studies evaluating its kinetics during tasks and the use of other related, but distinct perceptual responses (e.g. perceived effort; Halperin & Emanuel, (2020); Steele, (2020)) as a surrogate for the subjective experience of fatigue. 

Regarding your second question, this is an important point. We followed the authors of the RoF scale instructions and defined fatigue as “a feeling of diminishing capacity to cope with physical or mental stressors, either imagined or real” (Micklewright et al., 2017). This definition indicates that participants were to rate a global perceptual state, encompassing physical and cognitive components. Through general discussions with the participants, we believed that they all understood that they were rating their global, whole-body perceptual state, not just local sensory cues, though we did not directly quantify that. It is possible that participants may differentiate RoF and focus primarily on local sensory cues when rating RoF rather than a general feeling state. However, that is a research question outside of the scope of the presented manuscript. We believe this is unlikely.

6. Why did the authors decide to use this specific fatigue scale and not a visual analog scale of fatigue or another tool classically used to investigate fatigue? A better justification of the choice of this scale to investigate fatigue could help the reader.

There still remains no standard, universally-accepted instrument used for the assessment of the perception of fatigue (Dittner et al., 2004). As such, adopted measures should consider what aspects of fatigue are of interest and whether the instrument is suitable given both the population and experimental design (Dittner et al., 2004). Here we were interested in capturing perceived fatigue at the end of physical tasks and during physical tasks, alongside other perceptual and affective tools. The use of multi-item instruments (e.g. POMS) may prove too slow for this experimental design (particularly when assessing response during tasks), limiting acute responses from being captured (Micklewright et al., 2017). Visual analog scales (VAS) may hold some validity in the assessment of perceived fatigue but the results are highly dependent upon how the scale is presented to participants. Simply asking participants to indicate how fatigued they feel is not sufficient (e.g. participants may not be able to distinguish between perceived fatigue and sleepiness; see Dittner et al., 2004) and thus an explicit definition of fatigue is required (which itself also presents difficulty). There are also concerns about response bias with VAS (possibly reflecting a reluctance of participants to use the highest and lowest extremes of the scale). We believed that the RoF presented a suitable option for the aims of the study. It is a validated scale that is separable from other perceptual responses utilised within the study (i.e. perceived effort) (Micklewright et al., 2017). Some may have concerns about the definition used, believing it to be to close to self-efficacy, however as stated recent conceptualisations of fatigue identify reduced self-efficacy with the experience of fatigue (Stephan et al., 2016; Steele, 2021). Research utilising the RoF is still in its infancy and outstanding questions (a few highlighted above) still remain. Future research is required to investigate the psychometric properties of the RoF scale as a measure of perceived fatigue, particularly within a clinical context. This research appears forth coming, with its recent use in monitoring fatigue within clinical exercise programmes (Twomey et al., 2018).

We have added a brief rationale for the use of the RoF in the materials and methods section:

Lines 217-221: “The scale has been shown to have good face validity and high divergent validity from other related, but distinct, perceptual constructs (e.g. perceived effort) [11]. Therefore the scale was adopted to capture the dynamic experience of fatigue arising during the performance of a task that may be missed with the use of other multi-item instruments [11].”

SPECIFIC COMMENTS

- Abstract line 5: “perceptual (effort)”. Please indicate perception of effort.

This has now been changed. 

- L22-25: I respectfully disagree with this statement. As presented in point 4 above, the literature on the interaction between mental fatigue and physical performance extensively monitored changes in perception of fatigue induced by prolonged mental exertion and its effect on physical endurance.

The literature highlighted does not explicitly demonstrate that a perceived state of fatigue (induced by the cognitive tasks) per se influences subsequent physical performance. It is important to recognise that current perspectives on “mental fatigue” hold that it may manifest subjectively, behaviourally and/or physiologically (Van Cutsem et al., 2017). Van Cutsem et al., (2017) highlight in their review that only in six out of the eleven studies reviewed did a change in perceived fatigue occur with the performance of prior cognitive task (irrespective of whether performance deficits ensued in the subsequent physical task). So what is typically referenced as “mental fatigue” encompasses changes beyond just changes in the subjective experience of fatigue. 

Even in the studies that do demonstrate an increased perceived fatigue, these studies (on the whole) do not establish the relationship between a state of perceived fatigue and effort (from what we can tell is often just assumed, probably as a response to the vagilities of what mental fatigue represents) – an increase in perceived fatigue following the cognitive task cannot just be assumed to hold a causal role in processes leading to a subsequent elevation in perceived effort from such studies, as the effect on perceived fatigue from the performance of the cognitive task cannot be disassociated from the performance of the cognitive task itself (and thus any other effects of this, that are separate from perceived fatigue). As we have highlighted, only a few studies have identified and explicitly assessed the relationship between perceived fatigue and effort (Benoit et al., 2019; Greenhouse-Tucknott, Wrightson, et al., 2020; Harris & Bray, 2019), but these correlations again do not establish causality. That is why we conducted the present study; to see whether specifically perceived fatigue plays an important role in how effortful we perceive tasks to be. 

To try and be clear to what we are referring to we have amended this part of the introduction to make our point clearer

Lines 32-37: “The development of a perceived state of fatigue may similarly limit physical endurance performance by also influencing similar sensory processes [29]. Yet to the best of our knowledge, direct assessment of the relationships between perceptual and affective constructs is currently limited to a small number of correlation-based investigations in healthy populations across both physical and cognitive domains [29–31].”

- L36: the use of the reference Inzlicht et al. 2018 The effort paradox to justify the statement on perceived fatigue is not appropriate as the reference Inzlicht focuses on effort and not fatigue. Please use another reference to avoid confusing the reader between the constructs fatigue and effort.

We understand that. We cited Inzlicht and colleagues (2018) paper because their speculation on the shape of the effort-value relationship influenced our thinking concerning the shape of the fatigue-effort/performance relationship. We have amended the reference to make sure the reader is aware of the content of the original article. 

Lines 48-51: “Alternatively, an increase in perceived fatigue may evoke some change in these regulatory responses, but further increments in subjective intensity elicit no further changes (of note, similar considerations have recently been posed of the effort-value relationship [32]).”

- L80-83: thank you for this clear justification of the sample size.

Thank you for acknowledging this. 

- L114-116: thank you for this information. It is nice to see such detail being considered in the experimental design, i.e., influence of encouragement on performance.

Thank you for your kind words.

- L160. “Apparatus and procedures”: please add the information in the main manuscript and not in supplemental material. Such information is crucial for the reader to understand what was done, and the reader should not have to access to a supplemental material to obtain this information when the journal does not impose a word limit. Is a word limit imposed by PloS ONE?

This has been amended, with the description of the experimental set-up and apparatus used added back into the material and methods. 

References

Aboodarda, S. J., Iannetta, D., Emami, N., Varesco, G., Murias, J. M. and Millet, G. Y. (2020). Effects of pre-induced fatigue vs. concurrent pain on exercise tolerance, neuromuscular performance and corticospinal responses of locomotor muscles. Journal of Physiology, 598: 285–302.

Ainley, V. L., Tajadura-Jiménez, A., Fotopoulou, A., and Tsakiris, M. (2012). Looking into myself: Changes in interoceptive sensitivity during mirror self-observation. Psychophysiology, 49(11), 1672–1676. https://doi.org/10.1111/j.1469-8986.2012.01468.x

Ainley, V. L., and Tsakiris, M. (2013). Body Conscious? Interoceptive Awareness, Measured by Heartbeat Perception, Is Negatively Correlated with Self-Objectification. PLoS ONE, 8(2). https://doi.org/10.1371/journal.pone.0055568

Amann, M., Venturelli, M., Ives, S., McDaniel, J., Layec, G., Rossman, M. J. and Richardson, R. S. (2013). Peripheral fatigue limits endurance exercise via a sensory feedback-mediated reduction in spinal motoneuronal output. Journal of Applied Physiology (on-line), 115: 355–364. http://www.ncbi.nlm.nih.gov/pubmed/23722705.

Avery, J. A., Drevets, W. C., Moseman, S. E., Bodurka, J., Barcalow, J. C., and Simmons, W. K. (2014). Major depressive disorder is associated with abnormal interoceptive activity and functional connectivity in the insula. Biological Psychiatry, 76(3), 258–266. https://doi.org/10.1016/j.biopsych.2013.11.027

Avery, J. A., Kerr, K. L., Ingeholm, J. E., Burrows, K., Bodurka, J., and Simmons, W. K. (2015). A common gustatory and interoceptive representation in the human mid-insula. Human Brain Mapping, 36(8), 2996–3006. https://doi.org/10.1002/hbm.22823

Bandura, A. (1997). Self-efficacy: The exercise of control. New York: W.H. Freeman and Company.

Behm, D. G., Alizadeh, S., Anvar, S. H., Hanlon, C., Ramsay, E., Mahmoud, M. M. I., Whitten, J., Fisher, J. P., Prieske, O., Chaabene, H., Granacher, U. and Steele, J. (2021). Non-local muscle fatigue effects on muscle strength, power, and endurance in healthy individuals: A systematic review and meta-analysis. Sports Medicine, 51: 1893–1907.

Benoit, C. E., Solopchuk, O., Borragán, G., Carbonnelle, A., Van Durme, S. and Zénon, A. (2019). Cognitive task avoidance correlates with fatigue-induced performance decrement but not with subjective fatigue. Neuropsychologia (on-line), 123: 30–40. https://doi.org/10.1016/j.neuropsychologia.2018.06.017.

Bornemann, B., and Singer, T. (2017). Taking time to feel our body: Steady increases in heartbeat perception accuracy and decreases in alexithymia over 9 months of contemplative mental training. Psychophysiology, 54(3), 469–482. https://doi.org/10.1111/psyp.12790

Brown, D. M. Y., Graham, J. D., Innes, K. I., Harris, S., Flemington, A. and Bray, S. R. (2020). Effects of prior cognitive exertion on physical performance: A systematic review and meta-analysis. Sports Medicine, 50: 497–529.

Cameron, O. G. (2001). Interoception : The Inside Story — A Model for Psychosomatic Processes. Psychosomatic Medicine, 710, 697–710.

Van Cutsem, J., Marcora, S. M., De Pauw, K., Bailey, S. J., Meeusen, R. and Roelands, B. (2017). The effects of mental fatigue on physical performance: A systematic review. Sports Medicine, 47: 1569–1588.

Dittner, A. J., Wessely, S. C. and Brown, R. G. (2004). The assessment of fatigue: A practical guide for clinicians and researchers. Journal of Psychosomatic Research, 56: 157–170.

Durlik, C., Brown, G., and Tsakiris, M. (2014). Enhanced interoceptive awareness during anticipation of public speaking is associated with fear of negative evaluation. Cognition and Emotion, 28(3), 530–540. https://doi.org/10.1080/02699931.2013.832654

Farb, N. A. S., Segal, Z. V., and Anderson, A. K. (2013). Mindfulness meditation training alters cortical representations of interoceptive attention. Social Cognitive and Affective Neuroscience, 8(1), 15–26. https://doi.org/10.1093/scan/nss066

Garfinkel, S. N., and Critchley, H. D. (2013). Interoception, emotion and brain: new insights link internal physiology to social behaviour. Commentary on:: “Anterior insular cortex mediates bodily sensibility and social anxiety” by Terasawa et al. (2012). Social Cognitive and Affective Neuroscience, 8(3), 231–234. https://doi.org/10.1093/scan/nss140

Garfinkel, S. N., Manassei, M. F., Hamilton-Fletcher, G., den Bosch, Y. I., Critchley, H. D., and Engles, M. (2016). Interoceptive dimensions across cardiacand respiratory axes. Philosophical Transactions of the Royal Society B, 371, 20160014. https://doi.org/x.doi.org/10.1098/rstb.2016.0014

Garfinkel, S. N., Seth, A. K., Barrett, A. B., Suzuki, K., and Critchley, H. D. (2015). Knowing your own heart: Distinguishing interoceptive accuracy from interoceptive awareness. Biological Psychology, 104, 65–74. https://doi.org/10.1016/j.biopsycho.2014.11.004

Greenhouse-Tucknott, A., Butterworth, J. B., Wrightson, J. G., Smeeton, N. J., Critchley, H. D., Dekerle, J. and Harrison, N. A. (2020). Towards the unity of pathological and exertional fatigue: A predictive processing model.

Greenhouse-Tucknott, A., Pickering, S., Butterworth, J., Smeeton, N., Wrightson, J. and Dekerle, J. (2021). Prolonged cognitive activity increases perception of fatigue but does not influence perception of effort, affective valence, or performance during subsequent isometric endurance exercise. Sport, Exercise and Performance Psychology.

Greenhouse-Tucknott, A., Wrightson, J. G., Raynsford, M., Harrison, N. A. and Dekerle, J. (2020). Interactions between perceptions of fatigue, effort and affect decrease knee extensor endurance performance following upper body motor activity, independent of changes to neuromuscular function. Psychophysiology, 57: e13602.

Georgiou, E., Matthias, E., Kobel, S., Kettner, S., Dreyhaupt, J., Steinacker, J. M., and Pollatos, O. (2015). Interaction of physical activity and interoception in children. Frontiers in Psychology, 6(APR), 1–8. https://doi.org/10.3389/fpsyg.2015.00502

Halperin, I., Chapman, D. W. and Behm, D. G. (2015). Non-local muscle fatigue: Effects and possible mechanisms. European Journal of Applied Physiology, 115: 2031–2048.

Halperin, I. and Emanuel, A. (2020). Rating of perceived effort: Methodological concerns and future directions. Sports Medicine (on-line), 50: 679–687. https://doi.org/10.1007/s40279-019-01229-z.

Harris, S. and Bray, S. R. (2019). Effects of mental fatigue on exercise decision-making. Psychology of Sport and Exercise (on-line), 44: 1–8. https://doi.org/10.1016/j.psychsport.2019.04.005.

Harrison, N. A., Gray, M. A., Gianaros, P. J., and Critchley, H. D. (2010). The Embodiment of Emotional Feelings in the Brain. Journal of Neuroscience, 30(38), 12878–12884. https://doi.org/10.1523/JNEUROSCI.1725-10.2010

Harver, A., Katkin, E. S., and Bloch, E. (1993). Signal-detection outcomes on heartbeat and respiratory resistance detection tasks in male and female subjects. Psychophysiology, 30(3), 223–230. https://doi.org/http://dx.doi.org/10.1111/j.1469-8986.1993.tb03347.x

Herbert, B. M., Muth, E. R., Pollatos, O., and Herbert, C. (2012). Interoception across modalities: On the relationship between cardiac awareness and the sensitivity for gastric functions. PLoS ONE, 7(5), 1–9. https://doi.org/10.1371/journal.pone.0036646

Hockey, R. (2013). The psychology of fatigue: Work, effort and control. Cambridge University Press.

Holgado, D., Sanabria, D., Perales, J. C. and Vadillo, M. A. (2020). Mental fatigue might be not so bad for exercise performance after all: A systematic review and bias-sensitive meta-analysis. Journal of Cognition, 3: 38.

Holgado, D., Troya, E., Perales, J. C., Vadillo, M. A. and Sanabria, D. (2020). Does mental fatigue impair physical performance? A replication study. European Journal of Sport Science.

Inzlicht, M., Shenhav, A. and Olivola, C. Y. (2018). The effort paradox: Effort is both costly and valued. Trends in Cognitive Sciences, 22: 337–349.

Johnson, M. A., Sharpe, G. R., Williams, N. C. and Hannah, R. (2015). Locomotor muscle fatigue is not critically regulated after prior upper body exercise. Journal of Applied Physiology (on-line), 119: 840–850. http://jap.physiology.org/lookup/doi/10.1152/japplphysiol.00072.2015.

Jones, G., and Hollandsworth, J. (1981). Heart Rate Discrimination Before and After Exercise-Induced Augmented Cardiac Activity. Psychophysiology, 18(3), 252–257. https://doi.org/10.1111/j.1469-8986.1981.tb03029.x

Khalsa, S. S., Adolphs, R., Cameron, O. G., Critchley, H. D., Davenport, P. W., Feinstein, J. S., … Zucker, N. (2018). Interoception and Mental Health: A Roadmap. Biological Psychiatry: Cognitive Neuroscience and Neuroimaging, 3(6), 501–513. https://doi.org/10.1016/j.bpsc.2017.12.004

Khalsa, S. S., Craske, M. G., Li, W., Vangala, S., Strober, M., and Feusner, J. D. (2015). Altered interoceptive awareness in anorexia nervosa: Effects of meal anticipation, consumption and bodily arousal. International Journal of Eating Disorders, 48(7), 889–897. https://doi.org/10.1002/eat.22387

Khalsa, S. S., Rudrauf, D., Hassanpour, M. S., Davidson, R. J., and Tranel, D. (2020). The practice of meditation is not associated with improved interoceptive awareness of the heartbeat. Psychophysiology, 57(2), 1–16. https://doi.org/10.1111/psyp.13479

Khalsa, S. S., Rudrauf, D., Sandesara, C., Olshansky, B., and Tranel, D. (2009). Bolus isoproterenol infusions provide a reliable method for assessing interoceptive awareness. International Journal of Psychophysiology, 72(1), 34–45. https://doi.org/10.1016/j.ijpsycho.2008.08.010

Kuppuswamy, A. (2017). The fatigue conundrum. Brain, 140: 2240–2245.

Leproult, R., Van Reeth, O., Byrne, M. M., Sturis, J. and Van Cauter, E. (1997). Sleepiness, performance, and neuroendocrine function during sleep deprivation: Effects of exposure to bright light or exercise. Journal of Biological Rhythms, 12: 245–258.

Malliani, A., Lombardi, F., and Pagani, M. (1986). Sensory innervation of the heart. Progress in Brain Research, 67(C), 39–48. https://doi.org/10.1016/S0079-6123(08)62755-7

Manjaly, Z. M., Harrison, N. A., Critchley, H. D., Do, C. T., Stefanics, G., Wenderoth, N., … Stephan, K. E. (2019). Pathophysiological and cognitive mechanisms of fatigue in multiple sclerosis. Journal of Neurology, Neurosurgery and Psychiatry, 90(6), 642–651. https://doi.org/10.1136/jnnp-2018-320050

Matsumoto, Y., Mishima, K., Satoh, K., Shimizu, T. and Hishikawa, Y. (2002). Physical activity increases the dissociation between subjective sleepiness and objective performance levels during extended wakefulness in human. Neuroscience Letters, 326: 133–136.

Mehling, W. E., Chesney, M. A., Metzler, T. J., Goldstein, L. A., Maguen, S., Geronimo, C., … Neylan, T. C. (2018). A 12-week integrative exercise program improves self-reported mindfulness and interoceptive awareness in war veterans with posttraumatic stress symptoms. Journal of Clinical Psychology, 74(4), 554–565. https://doi.org/10.1002/jclp.22549

Meyerholz, L., Irzinger, J., Witthöft, M., Gerlach, A. L., and Pohl, A. (2019). Contingent biofeedback outperforms other methods to enhance the accuracy of cardiac interoception: A comparison of short interventions. Journal of Behavior Therapy and Experimental Psychiatry, 63(March 2018), 12–20. https://doi.org/10.1016/j.jbtep.2018.12.002

Micklewright, D., St Clair Gibson, A., Gladwell, V. and Al Salman, A. (2017). Development and validity of the Rating-of-Fatigue scale. Sports Medicine, 47: 2375–2393.

Morgan, P. T., Bailey, S. J., Banks, R. A., Fulford, J., Vanhatalo, A. and Jones, A. M. (2019). Contralateral fatigue during severe-intensity single-leg exercise: Influence of acute acetaminophen ingestion. American Journal of Physiology - Regulatory, Integrative and Comparative Physiology, 317: R346–R354.

Müller, T. and Apps, M. (2019). Motivational fatigue: A neurocognitive framework for the impact of effortful exertion on subsequent motivation. Neuropsychologia (on-line), 123: 141–151. https://doi.org/10.1016/j.neuropsychologia.2018.04.030.

Pageaux, B., Marcora, S. M. and Lepers, R. (2013). Prolonged mental exertion does not alter neuromuscular function of the knee extensors. Medicine and Science in Sports and Exercise, 45: 2254–2264.

Perakakis, P., Luque-Casado, A., Ciria, L. F., Ivanov, P. C., and Sanabria, D. (2017). Neural Responses to Heartbeats of Physically Trained and Sedentary Young Adults. BioRxiv, 156802. https://doi.org/10.1101/15680

Raccuglia, M., Heyde, C., Lloyd, A., Ruiz, D., Hodder, S. and Havenith, G. (2018). Anchoring biases affect repeated scores of thermal, moisture, tactile and comfort sensations in transient conditions. International Journal of Biometeorology, 62: 1945–1954.

Rae, C. L., Larsson, D. E. O., Garfinkel, S. N., and Critchley, H. D. (2019). Dimensions of interoception predict premonitory urges and tic severity in Tourette syndrome. Psychiatry Research, 271(December 2018), 469–475. https://doi.org/10.1016/j.psychres.2018.12.036

Steele, J. (2021). What is (perception of) effort? Objective and subjective effort during task performance.

Stephan, K. E., Manjaly, Z. M., Mathys, C. D., Weber, L. A. E., Paliwal, S., Gard, T., Tittgemeyer, M., Fleming, S. M., Haker, H., Seth, A. K. and Petzschner, F. H. (2016). Allostatic self-efficacy: A metacognitive theory of dyshomeostasis-induced fatigue and depression. Frontiers in Human Neuroscience (on-line), 10: 550. http://journal.frontiersin.org/article/10.3389/fnhum.2016.00550/full.

Stewart, J. L., May, A. C., Poppa, T., Davenport, P. W., Tapert, S. F., and Paulus, M. P. (2014). You are the danger: Attenuated insula response in methamphetamine users during aversive interoceptive decision-making. Drug and Alcohol Dependence, 142, 110–119. https://doi.org/10.1016/j.drugalcdep.2014.06.003

Twomey, R., Martin, T., Temesi, J., Culos-Reed, S. N. and Millet, G. Y. (2018). Tailored exercise interventions to reduce fatigue in cancer survivors: Study protocol of a randomized controlled trial. BMC Cancer, 18: 757.

Vaitl, D. (1996). Interoception. Biological Psychology, 42(1–2), 1–27. https://doi.org/10.1016/0301-0511(95)05144-9

Wallman-Jones, A., Perakakis, P., Tsakiris, M., and Schmidt, M. (2021). Physical activity and interoceptive processing: Theoretical considerations for future research. International Journal of Psychophysiology, 166(April), 38–49. https://doi.org/10.1016/j.ijpsycho.2021.05.002

Whitehead, W. E., and Drescher, V. M. (1980). Perception of gastric contractions and self-control of gastric motility. Psychophysiology, 17(6), 552–558.

Zarza, J. A., Sanabria, D., and Perakakis, P. (2019). Can increased interoception explain exercise-induced benefits on brain function and cognitive performance? Experimental Psychology, 1–16.

Zoellner, L. a., and Craske, M. G. (1999). Interoceptive accuracy and panic. Behaviour Research and Therapy, 37(12), 1141–1158. https://doi.org/10.1016/S0005-7967(98)00202-2

---

## [Decision Letter · Decision Letter 1]

21 Dec 2021

Effect of the subjective intensity of fatigue and interoception on perceptual regulation and performance during sustained physical activity

PONE-D-21-15950R1

Dear Mr  Aaron Greenhouse-Tucknott,

We’re pleased to inform you that your manuscript has been judged scientifically suitable for publication and will be formally accepted for publication once it meets all outstanding technical requirements.

Kind regards,

Mathieu Gruet, Ph.D

Academic Editor

PLOS ONE

Additional Editor Comments (optional):

Reviewers' comments:

Reviewer's Responses to Questions

**Comments to the Author**

1. If the authors have adequately addressed your comments raised in a previous round of review and you feel that this manuscript is now acceptable for publication, you may indicate that here to bypass the “Comments to the Author” section, enter your conflict of interest statement in the “Confidential to Editor” section, and submit your "Accept" recommendation.

Reviewer #1: All comments have been addressed

Reviewer #2: (No Response)

Reviewer #3: All comments have been addressed

2. Is the manuscript technically sound, and do the data support the conclusions?

Reviewer #1: Yes

Reviewer #2: Yes

Reviewer #3: Yes

3. Has the statistical analysis been performed appropriately and rigorously? 

Reviewer #1: Yes

Reviewer #2: Yes

Reviewer #3: Yes

4. Have the authors made all data underlying the findings in their manuscript fully available?

Reviewer #1: Yes

Reviewer #2: Yes

Reviewer #3: Yes

5. Is the manuscript presented in an intelligible fashion and written in standard English?

Reviewer #1: Yes

Reviewer #2: Yes

Reviewer #3: Yes

6. Review Comments to the Author

Reviewer #1: I had only minor comments for this mansucritp, and all of them have been appropriately adressed.

I have no further comments.

Reviewer #2: Apologies again for my slowness to complete the re-review! They always seem to come back to my inbox at times when I am swamped. Anyway, I'm happy with all your responses to my comments. Great work! Thanks so much for engaging with my (sometimes) ramblings, and putting up with me causing issues in rerunning analyses etc.

I'll just add a quick clarification about Massin's comment, and what seems like a disagreement. You are right, his comment appears to suggest using 'perceived' rather than 'perception of', but in fact I opt for the latter precisely because I agree with him that 'perception of' an x of kind K is not necessarily itself of kind K. By this I mean that I am not entirely sure that our perception of effort is indeed really a perception of effort... at least a perception of the thing I define as (actual) effort. This is why I have an interest in exploring the psychophysics of this phenomenological experience more. Although, in re-reading Massin's comment I also think he may contradict himself somewhat too... the tree example is probably more akin to what I was getting at. The 'perceived' tree is just the tree that you perceive. It refers to the existent that you are representing, not the representation itself. Its sort of like saying the kicked ball... in this we just mean the ball that was kicked. For effort we could say perceived effort in which what we mean is the effort that was perceived. But if we are speaking about the phenomenology, then we care not about the effort actual, but the perception of that effort (or to use Nagel's or Dennett's language, the perception of what "seems" to be that effort - this is what I mean by being unsure of whether our perception of effort is really a perception of 'effort' as I define). Like I said originally though, it's not a big deal and I'm just being pedantic really... no need to change anything, just wanted to clarify my position ;-)

Reviewer #3: Thank you for your detailed answer to the reviewers comments as well as the updated manuscript.

I have no other comments related to the manuscript. You have answered/addressed all my comments. I would like to congratulate you again for this really nice study and manuscript, as well as your previous studies on this topic. Your studies are really interesting.

I have only two comments related to the response on mental fatigue. I think hese two points are more a brainstorming to share with you.

1. I would be cautious with the systematic review and meta-analysis of Holgado and colleagues when questioning the true existence of a mental fatigue effect. I would like to point that the original publication and associated effect size obtained was incorrect. The authors published a correction/erratum due to some errors in the calculation. When correcting these errors, they observed an effect =-0.53 with no confidence interval crossing zero (-0.76 ; -0.25). Such effect, from my point of view, is of great importance for performance (as well as other domains potentially impacted by mental fatigue such as cognitive performance or locomotion).

This being said, I fully agree with you that other replication studies are needed.

2. Another important limitation of this meta analysis is the lack of consideration that physical performanceS are differently regulated. Contrary to Pageaux and colleagues 2018 or Brown and colleagues 2020, Holgado and colleagues did not consider that endurance performance is differently regulated than maximal force production performance or psychomotor performance for example. Interestingly, when considering each performance separately it appears that the negative effect of mental fatigue on subsequent physical performance is confined to endurance and psychomotor performance, and not maximal force production capacity. Therefore, when pooling all performance altogether, it is likely that that the effect calculated will be underestimated.

This is I think an important consideration when considering how mental fatigue could impact physical performance in humans.

Thanks again for giving me the opportunity to review your manuscript. And congratulations for this really nice work.

7. PLOS authors have the option to publish the peer review history of their article (what does this mean?). If published, this will include your full peer review and any attached files.

Reviewer #1: **Yes: **Robin Souron

Reviewer #2: **Yes: **James Steele

Reviewer #3: No

---

## [Editor Report · Acceptance letter]

23 Dec 2021

PONE-D-21-15950R1 

Effect of the subjective intensity of fatigue and interoception on perceptual regulation and performance during sustained physical activity 

Dear Dr. Greenhouse-Tucknott:

I'm pleased to inform you that your manuscript has been deemed suitable for publication in PLOS ONE. Congratulations! Your manuscript is now with our production department. 

Kind regards, 

on behalf of

Dr. Mathieu Gruet 

Academic Editor

PLOS ONE